# How does Chain of Thought decompose complex tasks?

**Amrut Nadgir** [1]   **Vijay Balasubramanian** [* 1]   **Pratik Chaudhari** [* 1]

## Abstract

Many language tasks can be modeled as classification problems where a large language model (LLM) is given a prompt and selects one among many possible answers. We show that the classification error in such problems scales as a power law in the number of classes. This has a dramatic consequence: the prediction error can be reduced substantially by splitting the overall task into a sequence of smaller classification problems, each with the same number of classes ("degree"). This tree-structured decomposition models chain-of-thought (CoT). It has been observed that CoT-based predictors perform better when they "think", i.e., when they develop a deeper tree, thus decomposing the problem into a larger number of steps. We identify a critical threshold for the degree, below which thinking is detrimental, and above which there exists an optimal depth that minimizes the error. It is impossible to surpass this minimal error by increasing the depth of thinking.

## 1 Introduction

Training Large Language Models (LLMs) to mimic human reasoning via techniques such as Chain of Thought (CoT) and "thinking" has led to impressive advances in the ability of computers to carry out mathematical reasoning and programming (Wei et al., 2022; Lewkowycz et al., 2022; Li et al., 2022). On one hand, some studies have demonstrated that excessive thinking, where models generate long reasoning traces to solve problems, can hurt performance (Shojaee* et al., 2025; Wu et al., 2026; Liu et al., 2025b). On the other hand, DeepSeek-R1-Zero achieves excellent performance on mathematical reasoning benchmarks despite its tendency to construct long reasoning paths that seem convoluted to the human eye (Guo et al., 2025). Such conflicting observations suggest that naively increasing reasoning length might not lead to improvements. Our goal in

this paper is to propose criteria as to when and why reasoning works and how much of it a machine should do.

Consider a task where a large language model (LLM) is given a prompt and it selects an answer from many possible choices. This is, in effect, a classification problem. The prompt often consists of two pieces, a context containing background information and a question. The LLM could either produce the answer directly, or it could identify the answer using a sequence of steps; each successive step corresponding to a classification sub-task that effectively extends the context for the next step. The text associated with this sequence of classifications is called a "chain of thought". By construction, chain of thought has a tree-like structure. LLMs are often instructed to "think" by appending this chain of thought to the context iteratively to build extended reasoning traces. Thinking is a concatenation of multiple chains of thought, and therefore it also has a tree-like structure.

We first show that the probability of error in standard supervised learning classification problems scales as $m^{2/d}D^{-1/d}$ where $m$ is the number of classes, $D$ is the number of data points and $d$ is the intrinsic dimension of the input domain. Specifically, the error scales as a power law in the number of classes. This is a general result that holds for any learning or inference mechanism—including LLMs for which the number of classes equals the number of plausible answers. We then show that the classification error can be reduced substantially by decomposing the task into a sequence of smaller problems. The error is minimized when each sub-problem has a certain optimal degree $m^* = e^{d/2}$ (number of classes), i.e., the tree of decisions is balanced. This tree-structured decomposition describes the sequential production of tokens as an LLM uses chain of thought. We then show that for the same number of total classes, thinking, namely increasing the length of CoT, leads to deterioration of the error if the tree has a degree smaller than $m^*$. However, if the degree is bigger than $m^*$ then thinking reduces the error up to a certain depth. It is impossible to surpass the resulting minimum error by further increasing CoT length.

Our work provides a simple explanation for CoT by formalizing reasoning as the decomposition of a large classification task. This also explain empirical observations such as an optimal reasoning length (Wu et al., 2026) and the importance of structure in reasoning traces (Li et al., 2025).

---

[*]Equal contribution [1]University of Pennsylvania. Correspondence to: Amrut Nadgir <amrutn@sas.upenn.edu>.

*Proceedings of the 43rd International Conference on Machine Learning*, Seoul, South Korea. PMLR 306, 2026. Copyright 2026 by the author(s).

## 2 Classification error in supervised learning scales as a power law in the number of classes

We begin by identifying a scaling law for the probability of mis-classification as a function of the (i) number of data points, (ii) number of classes, and (iii) the dimensionality of the input space. Consider a dataset $\{(x_i, y_i)\}_{i=1}^{D}$ with $D$ samples. Inputs $x_i$ are drawn independently from the distribution $q(x)$ supported on the $d$-dimensional, finite-volume input domain $\mathcal{X}$, and outputs are discrete-valued $y_i \in \{1, \ldots, m\}$. Strictly speaking, the prompts in an LLM are supported on a discrete set, but for the purposes of this analysis we can think of $x_i$ as the embedding of a prompt into a vector space. For the kinds of data we focus on in this paper, there is only one correct output for every input, e.g., 2 + 3 * 4 + 5 = 19. The data used to train an LLM can contain mistakes. Nevertheless, we assume that the correct answer occurs most often in the data. Assume that each class has an equal number of samples $(D/m)$. [1] A classifier learns a probability distribution $p(y \mid x)$ using this dataset.

**We will first analyze the error of a probabilistic predictor.** Suppose we have a "well-trained" classifier so that, on the $D$ samples in the dataset, the learned distribution $p(y \mid x)$ is concentrated on the correct output class.[2] We would like to study the error, i.e., the probability of incorrect classification, for a well-trained classifier which samples the output from its learned distribution. Such probabilistic decoding, e.g., nucleus sampling (Holtzman et al., 2020), is commonly used when generating text from an LLM. The error is

$$E = 1 - \mathop{\mathbb{E}}_{x \sim q} \left[ p\left(y^*(x) \mid x\right) \right] \quad (1)$$

where $y^*(x)$ is the correct class corresponding to the input $x$. The error is minimized to zero when the learned distribution $p(y \mid x)$ concentrates all its probability on the correct class.

Empirical and theoretical results suggest that over-parameterized networks learn a smooth interpolation of observed data (Bubeck & Sellke, 2023; Bartlett et al., 2017), so there should be a limit on how quickly the learned distribution of the classifier changes with respect to its input. Let $K$ be the Lipschitz constant for $p(y \mid x)$, uniformly over all

---

[1]This essentially assumes that the training set has an equal number of questions for each possible answer. This is a reasonable assumption because in a sequence of such classification problems that we consider next, having a uniform marginal distribution over the outputs maximizes the mutual information of each decision with the final answer (MacKay, 2019, Chapter 4). If the marginal is not uniform, one might proceed in this analysis by approximating it as uniform over $m \sim \exp(H(y))$ classes where $H(y)$ denotes the Shannon entropy of the marginal over outputs $y$.

[2]These kinds of assumptions are common in the literature on non-parametric or over-parameterized estimators (Belkin et al., 2019; Zhang et al., 2017; Wahba, 1990) and have been used to explain neural scaling laws (Bahri et al., 2024).

$x$. The error is upper-bounded by

$$
\begin{aligned}
E &= 1 - \int_{\mathcal{X}} \mathrm{d}x \, q(x) p\left(y^*(x) \mid x\right) \\
&= \int_{\mathcal{X}} \mathrm{d}x \, q(x) \left[ p\left(y^*(x) \mid \hat{x}\right) - p\left(y^*(x) \mid x\right) \right] \quad (2) \\
&\leq K \int_{\mathcal{X}} \mathrm{d}x \, q(x) \|x - \hat{x}\|
\end{aligned}
$$

where $\hat{x}$ is the nearest point to $x$ in the training dataset such that $y^*(x) = y^*(\hat{x})$. In the above equation, the second line follows by adding and subtracting $p\left(y^*(x) \mid \hat{x}\right)$ to the integrand and using the fact that $p\left(y^*(x) \mid \hat{x}\right) = 1$. The third line applies the Lipschitz condition.

Let us average both sides of Eq. (2) over draws of the training data; let us denote this average by $\mathbb{E}[\cdot]$. The only quantity on the right-hand side that depends upon the dataset is $\hat{x}$. For a fixed $x \sim q$, the quantity $\mathbb{E}[\|x - \hat{x}\|]$ is the average distance between $x$ and $\hat{x}$ over draws of the dataset.

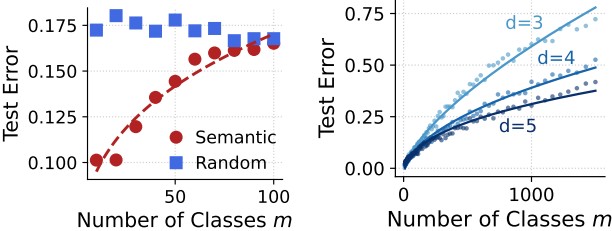

**Figure 1. Left:** Classification error of a vision transformer scales as a power law with the number of classes $m$ for semantic groupings (obtained by merging or splitting the original CIFAR-100 superclasses), but is roughly constant for random groupings. The fitted power law (dashed red) $0.036 \, m^{0.31} + 0.02$ corresponds to an intrinsic dimension of $d = 6.45$, which is close to the prior estimate of $d = 15 \pm 5$ by Pope et al. (2021). Random groupings don't obey the scaling law because they violate our assumption that each class $i$ corresponds to a continuous region $\mathcal{A}_i$ on the input space $\mathcal{X}$. Instead, each class corresponds to a union of disjoint regions. **Right:** This power-law trend in the test error is also evident in a student-teacher setting where we can vary both the number of classes $m$ and the dimensionality of the input space $d$. The lines are power-law fits of the form $a \, m^{2/d} + b$ with $a$ and $b$ fitted to the data and $d$ being the actual input dimensionality. Details for both experiments are in Section A.1.

Let us assume that the volume in $d$-dimensional input space corresponding to inputs $x$ corresponding to any class $y$ is $\sim V_0$ and is roughly isotropic. If we have equal amounts of data $D/m$ for each class, the typical separation of data within a class is $\sim (V_0/(D/m))^{1/d}$ which scales as $(D/m)^{-1/d}$. Thus, $\mathbb{E}[\|x - \hat{x}\|]$ for a fixed $y$ must scale as $(D/m)^{-1/d}$ with some constant of proportionality $\alpha$. The expected error from Eq. (2)

$$\bar{E} = \mathbb{E}[E] \leq \alpha K m^{1/d} D^{-1/d}.$$

Even if the volume $V_0$ varies across classes $y$, the scaling of the separation with respect to $d$ and $m$ will remain the same provided the training set has a similar number of samples $D/m$ for each class. The empirical results of Bahri et al. (2024) and our numerical experiment in Fig. 1 for the scaling laws of actual neural networks support the bounds derived.

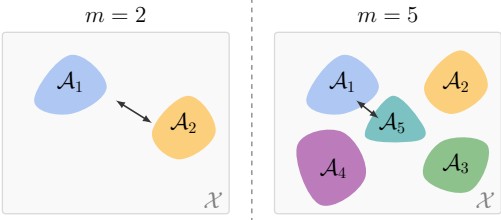

**Figure 2.** Illustration of regions $\mathcal{A}_i$ in the input domain $\mathcal{X}$ corresponding to different classes $i$. With fewer classes (left), regions are well-separated (long arrow). As the number of classes increases (right), regions are packed more densely which reduces the distance between them. This reduced margin forces the learned function to change rapidly, requiring a higher Lipschitz constant.

**We will next understand how the Lipschitz constant scales with the number of classes.** Fig. 2 summarizes the following argument. Consider the case where each class $i$ is associated with a continuous region $\mathcal{A}_i = \{x \in \mathcal{X} : p(i \mid x) > 1 - \epsilon\}$ for some constant $\epsilon > 0$. The constant $\epsilon$ is sufficiently small so that distinct regions do not overlap, i.e., $\mathcal{A}_i \cap \mathcal{A}_j = \emptyset$ if $i \neq j$, and sufficiently large so that $\mathcal{A}_i \neq \emptyset$ for any $i$. The minimum distance between two inputs in distinct regions $s_{ij} = \min_{x_i \in \mathcal{A}_i, x_j \in \mathcal{A}_j} \|x_i - x_j\|$ bounds the Lipschitz constant, $K \geq (1 - 2\epsilon)/s_{ij}$ for any $j \neq i$ and all $i$. If $s_{\min}$ is the minimum value of $s_{ij}$ over all distinct pairs $\mathcal{A}_i$ and $\mathcal{A}_j$, then $K \geq (1 - 2\epsilon)/s_{\min}$. For $m$ distinct classes, $s_{\min} = \mathcal{O}(m^{-1/d})$. This is because $s_{\min}$ is upper bounded by the maximum possible minimum pairwise distance between $m$ points in the input space. Altogether,

$$K = \Omega\left(m^{1/d}\right). \tag{3}$$

If the scaling of the Lipschitz constant with respect to $m$ matches this lower bound, which would give the most stringent bound on the error, we have,

$$\bar{E} = \mathcal{O}\left(m^{2/d} D^{-1/d}\right). \tag{4}$$

Hestness et al. (2017) observed a similar power-law for the test error as a function of the number of samples per class. Fig. 1 shows that the test error on CIFAR-100 classification tasks as well as on synthetic classification tasks (across different input dimensions $d$), indeed scales with respect to $m$ as the power-law in Eq. (4).

**Remark 1 (An analysis of greedy decoding).** Greedy decoding in an LLM chooses the most likely token given the context. Thus, the distribution that the classifier samples its outputs from is different from its learned distribution $p(y \mid x)$. Our previous argument undergoes minor modifications in this case. But due to the assumption that $p(y \mid \hat{x})$ concentrates on the correct class for $\hat{x}$ in the training data, greedy decoding for an input $x$ incurs an error only if the nearest training sample $\hat{x}$ with the same class satisfies $\|x - \hat{x}\| \gtrsim 1/K$. When we average $\|x - \hat{x}\|$ over the training dataset, the probability that we incur an error for $x$ is

$$\mathbb{P}(\|x - \hat{x}\| \gtrsim 1/K) \leq K \, \mathbb{E}\left[\|x - \hat{x}\|\right]$$

by Markov's inequality. The right-hand side here scales similarly to the one in Eq. (2) and therefore our scaling law for the error holds even for greedy decoding.

## 3 CoT works best when the reasoning tree has equal degree at each level. There is an optimal degree that maximizes accuracy.

We will next specialize the preceding argument for predictors that use Chain-of-Thought (CoT). As discussed in the introduction, CoT can be thought of as a sequence of classification problems. We can therefore use results from Section 2 to bound the error of this sequence of predictions, and compare it to the error of directly predicting the answer without CoT.

Consider an LLM that is trained to predict the next token $x_{k+1}$ from its past context $x_{0:k} = (x_0, \ldots, x_k)$. Let the first token $x_0$ be the prompt and $x_n$, the final token, be the answer selected from $N$ distinct possibilities for the task at hand. If the model directly predicts the answer from the prompt, i.e., if $n = 1$, our results in Section 2 apply directly. The expected error of such a predictor is

$$\bar{E}_{\text{direct}} = cN^{2/d} D^{-1/d} \tag{5}$$

where $d$ is the intrinsic dimension of the input prompt.

**Dimension $d$ of the input in an LLM** In Section 2, inputs were drawn from a $d$-dimensional space. In an LLM, the inputs are sentences. We want to estimate the dimension of these sentences. Although each token is represented as a vector in some, say, $p$-dimensional space, the set of such vectors that make up a sentence of length $n$ may lie within a subspace of lower dimensionality—which we will call the intrinsic dimension $d$. In Fig. 3, we estimated $d$ as the number of principal components required to capture 80% of the variance of the log-probabilities over vocabulary (logits) used to select the next token from the context. This is a common procedure Soatto et al. (2024); Hao et al. (2025); Azaria & Mitchell (2023). We can think of $d$ as the dimensionality of the latent representation that compresses the input sentence to produce the next token. Fig. 3 (left) shows that the intrinsic dimensionality of the latent state of an LLM is relatively stable as a function of the context length.

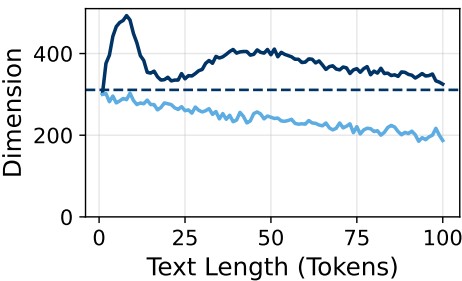 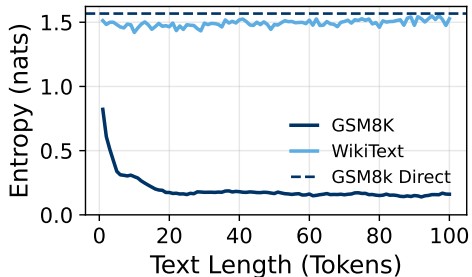

**Figure 3. Intrinsic dimension of the sequence (left) and mean entropy of the predicted next-token distribution (right)** for Qwen3-32B (Yang et al., 2025a) computed using 4,407 samples from GSM8k (Cobbe et al., 2021) (dark blue) with answers that have at least 100 tokens. For WikiText-2 (Merity et al., 2016) (light blue) we use 3,045 samples, which are at least 167 tokens long, The X-axis represents the length of the sequence. For GSM8k we do not include the length of the prompt in the plot (although the model has the prompt in its context when it produces the answer). For WikiText-2, we treat the first 67 tokens as the "prompt" to horizontally align the curves. The "GSM8k Direct" baseline (dashed blue) forces the model to output the answer immediately by adding the text "Only output the answer." to the prompt (with dimension and entropy computed using the first generated token, i.e., the answer). **Left:** Intrinsic dimension of the latent space $d$ (number of dimensions of principal components analysis that capture 80% of the variance) is similar for both mathematical tasks (GSM8k and GSM8k Direct) relative to a generic baseline (WikiText). The dimensionality $d$ is relatively constant with the text length. Section A.2 shows the same result with other, non-linear, dimensionality reduction methods. **Right:** Mean entropy of predicted next-token is small for GSM8k compared to predicting the answer directly (GSM8k-Direct). This shows that the model is more confident when predicting on human-generated reasoning traces, as compared to predicting the answers directly. The mean entropy of the predicted next token in reasoning traces of GSM8k (dark blue) is lower than that of WikiText (light blue), indicating that a well-trained model (Qwen3-32B) predicts more confidently on the former. Direct prediction of the answer in a reasoning task (GSM8k-Direct) is as unconstrained as predicting the next token in English text (WikiText), as indicated by the similarity of their entropies.

**The error of CoT-based predictions**  Next, consider the case where the model first generates a CoT before producing the final answer, i.e., $n > 1$. At a step $k$, the model has a choice of, say, $m_k$ tokens that it may produce with probability exceeding some threshold. CoT in an LLM therefore instantiates $n$ classification problems with $\{m_1, \ldots, m_n\}$ classes respectively.

**Remark 2** (**Why isn't the number of plausible next tokens equal to the vocabulary size?**). As Fig. 3 (right) shows, the number of plausible next tokens in a well-trained LLM when conditioned on the context is much smaller than the vocabulary size, for both natural language data like that in WikiText as well as mathematical reasoning traces in GSM8k. In fact, the entropy of the next token for reasoning traces in GSM8k is $\sim 10\times$ smaller than that of WikiText. This suggests that the context in reasoning tasks enormously constrains the number of plausible next tokens *at each step* of the reasoning trace. This is why, in our analysis we can assume that the number of classes in each classification problem $\{m_1, \ldots, m_k\}$ is much smaller than the vocabulary size.

Let us first consider the case where no two chains of thought lead to the same final token (answer). Then, the product of the degrees in the reasoning tree is equal to the number of leaves $\prod_{k=1}^{n} m_k = N$. Note that once an LLM is trained, the same network is used in an auto-regressive fashion to produce successive tokens along the reasoning trace. While the specific tokens corresponding to each of these $m_k$ options at level $k$ of the tree depends on the CoT history, the

underlying classification task is shared across branches of the reasoning tree and is learned from $D$ samples in the training dataset. In addition, Fig. 3 (left) shows that the dimension $d$ of the latent states of an LLM is relatively stable with respect to $k$ (dark blue) and has a similar value as the dimension for direct prediction (dashed blue). By the union bound, the probability of error is smaller than the sum of the probabilities of error at each reasoning step,

$$\bar{E}_{\text{reason}} \leq cD^{-1/d} \sum_{k=1}^{n} m_k^{2/d}. \qquad (6)$$

The difference in expected error between reasoning-based prediction and direct prediction

$$\bar{E}_{\text{direct}} - \bar{E}_{\text{reason}} \geq cD^{-1/d} \left[ \prod_{k=1}^{n} m_k^{2/d} - \sum_{k=1}^{n} m_k^{2/d} \right] \qquad (7)$$

where we have used the fact that the number of outputs is the product of the degrees, $N = \prod_{k=1}^{n} m_k$. CoT-based prediction reduces the error compared to direct prediction when the product of the degrees is greater than the sum.

**An analysis of when CoT is most effective**  To maximize the "reasoning gain", we need to minimize the sum $\sum_{k=1}^{n} m_k^{2/d}$ while keeping the product of the summands $N = \prod_{k=1}^{n} m_k^{2/d}$ fixed. The solution is easily shown, by the method of Lagrange multipliers, to require that all the degrees of the tree $m_k$ are equal. In other words, the reasoning tree is "maximally structured". Now observe that the

right-hand side of Eq. (7) is proportional to

$$\prod_{k=1}^{n} m_k^{2/d} - \sum_{k=1}^{n} m_k^{2/d} = N^{2/d} - \frac{\ln N}{\ln m} m^{2/d}$$

where we have substituted $n = \ln N / \ln m$ for fixed task size $N$ and constant degree $m$. This difference is maximized at

$$m^* = e^{d/2} \tag{8}$$

where $e$ is the base of the natural logarithm and the maximal reasoning gain is proportional to

$$\text{Optimal Reasoning Gain} = N^{2/d} - \frac{e \ln N}{d/2}. \tag{9}$$

This calculation is similar to the analysis of the organization of grid-cell modules in the brain for spatial navigation (Wei et al., 2015). Our results suggest that we could view of the hierarchy of grid-cell scales as providing a "reasoning trace" for self-localization.

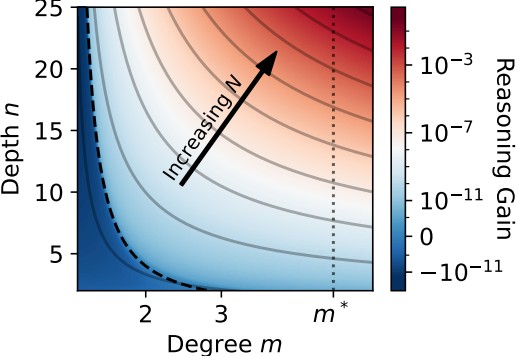

**Figure 4. Theoretical reasoning gain as a function of the degree $m$ and depth $n$, computed using Eq. (7).** Non-integer values of $m$ and $n$ are shown for completeness. The dimension is $d = 3$ and we set $cD^{-1/d} = 10^{-12}$ for the visualization. The black dashed curve is the boundary above which gain is positive. Gray curves are contours of the number of leaves $N = m^n$ (answers). The vertical dotted line marks the optimal degree $m^* = e^{d/2} \approx 4.5$, which maximizes reasoning gain given a task of a fixed size $N = \text{const}$.

When the tree has an equal degree at each level ($m_k = m$ for all $k$), reasoning is only beneficial if the task is sufficiently large, i.e., $N = m^n$ is high. Reasoning is detrimental on tasks with a small $N$ because the right-hand side of Eq. (7) is negative. This is also seen in Fig. 4. This corroborates prior work showing that reasoning can degrade performance on simple tasks (Shojaee* et al., 2025).

**Remark 3** (**Difficult problems are not necessarily the ones with a large reasoning depth**). The maximal gain on task with CoT in Eq. (9) compared to direct prediction depends only on the task size $N$ and intrinsic dimension $d$. Therefore, the difficulty of a task is not necessarily characterized by the length of reasoning traces—there is no $n$ in

the right-hand side of Eq. (9). This differs from claims made in the literature (Wu et al., 2026; Shojaee* et al., 2025; de Varda et al., 2025; Shen et al., 2025).

**Remark 4** (**The performance of LLMs is consistent with our theoretical predictions**). Our mathematical argument suggests that CoT-based prediction performs better than direct prediction because it decomposes a difficult classification task (large $N$) into a sequence of smaller, easier, ones. If LLMs are in fact described by this theory, then the uncertainty of the LLM while predicting the next token along "correct" reasoning traces would be lower than that of directly predicting the answer. Indeed, as we saw in Fig. 3 (right), the next-token predictions on correct reasoning traces have a low entropy compared to direct predictions of the answer. This is in spite of the fact that the dimensionality of the latent space dimensionality for the two cases is similar (Fig. 3, left).

**An experiment with synthetic data** Consider a problem where we would like to learn a map $\varphi : \mathbb{R}^d \times \{1, \ldots, N\} \mapsto \{0, 1\}$. The real-valued part of the input argument $v \in \mathbb{R}^d$ to $\varphi(v, k)$ models the context and the integer part $k$ models the prompt to an LLM. Suppose $\varphi$ has the structure depicted in Fig. 5 (left). The boolean value of the root (shaded circle) is determined by the dot product of $v$ with a fixed reference vector $w$. This value propagates down the tree according to the fixed logical operations, identity or negation ($\neg$), indicated on the edges. For example, if $\mathbf{1}\{v \cdot w > 0\}$ equals 1, then $x_1 = 1$ and $x_2 = \neg x_0 = 0$. Likewise $x_3 = \neg x_1 = 0$ and $x_4 = x_1 = 1$, and so on. A "direct" predictor of $\varphi(v, k)$ should learn the values of each leaf and produce the correct value for leaf, say $k = 5$, given the context $v$. A CoT-based predictor would, for example, predict the value of each node along the highlighted red path in the tree before predicting the value of the leaf $x_5$. This task design creates learnable patterns where a continuous set of inputs $v$ can be mapped to the values at the leaves. Due this structure, this problem can be either learned directly or by CoT that exploits the underlying structure. Section A.3 provides more details of the task and how we train a transformer to perform it.

Fig. 5 (middle) shows that transformers trained, using the standard next token prediction objective, on data from this task with different degrees and depths of the tree exhibit the smallest error when the degree of the tree is the same at each layer ("structure" equal to 1). This confirms the prediction from Eq. (9) that a balanced tree minimizes the error of CoT. Our experiment in Fig. 5 (middle) trains a causal self-attention-based transformer on the embedding of the context $v$ and that of the prompt, e.g., "Target X5". We tokenized the text in the dataset for this task using standard byte-pair encoding. We checked whether the ground-truth degree of the task is consistent with the apparent degree of the CoT reasoning traces after tokenization. This is interesting to do

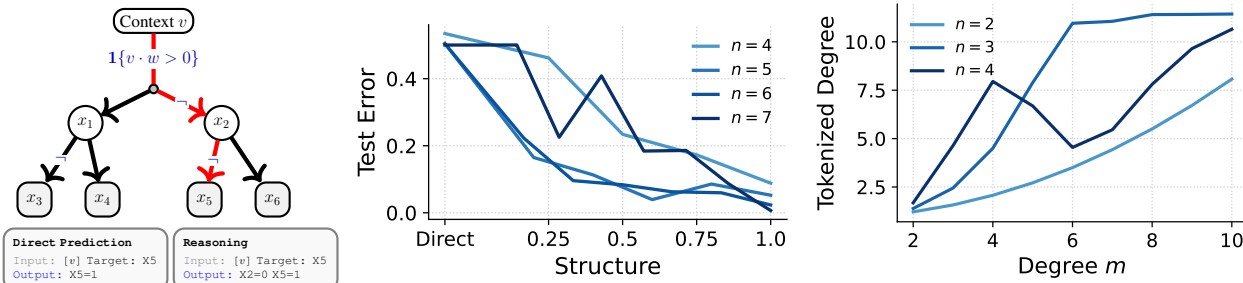

**Figure 5. Left:** A synthetic task where data exhibits a tree structure, see the narrative for a description. **Middle:** Transformers trained, using the standard next token prediction objective, on such tasks with different degrees and depths of the tree exhibit the smallest error when the degree of the tree is the same at each layer ("structure" equal to 1). Each curve corresponds to a tree of a fixed depth $n$ and degree 3, i.e., $3^n$ leaves. A structure equal to 1.0 corresponds to a tree with a constant degree of $m = 3$ across layers, while smaller values correspond to trees whose degree is different at different depths. **Right:** The degree of our synthetic task (x-axis) is positively correlated with the degree of a prefix-tree of tokenized reasoning traces (y-axis). To create the prefix-tree, we trained a byte-pair encoding tokenizer with a vocabulary size of 500 on all possible reasoning traces.

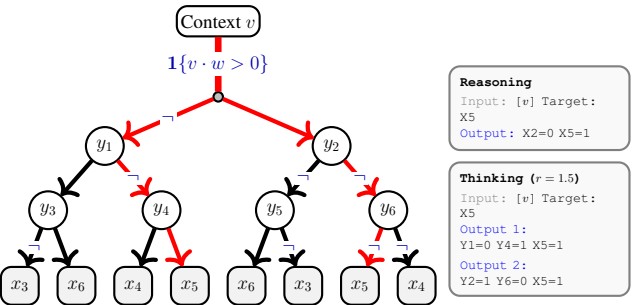

**Figure 6.** A "thinking" tree corresponding to the one in Fig. 5 (left) with increased depth from 2 to 3 and redundant leaves. Intermediate nodes in this tree are labeled with "Y" instead of "X". Each leaf in the original tree in Fig. 5 (left) is associated with two leaves in this deeper tree. The identity and negation ($\neg$) operations on the edges are chosen so that the binary values of leaves are consistent between both trees. The two reasoning paths in the thinking tree corresponding to the leaf $x_5$ in the original tree are highlighted in red. Either path can be used to deduce the correct value of $x_5$. The "Thinking Mode" box on the right has examples of chains of thought corresponding to both paths.

because the larger the degree of the task after tokenization, the larger the number of outputs at each level of the CoT tree that the LLM must resolve. Fig. 5 (right) shows the two are correlated. This finding validates the problem setup of our paper: the sequence of next-token predictions in an LLM is, in effect, a branch of a tree. The degree of this underlying tree is small—and the task amenable to CoT-based prediction—if the degree of the task is small.

## 4 There is a threshold for the degree, below which extended reasoning ("thinking") is detrimental, and above which there is an optimal depth that maximizes accuracy.

In real applications of LLMs there are often multiple reasoning paths to get to the same correct answer. To model this, we created the task depicted in Fig. 6 where data are

generated as described in Fig. 5 but where there are multiple leaves that consistently determine the same variable. For example, the two paths marked in red in Fig. 6 produce the value of the same variable $x_5$ given the context $v$. Such a tree models redundant reasoning paths.

Consider a task with $N = m^n$ possible answers that is represented by a tree of depth $n$ and degree $m$. Compare this task to another which has the same number of distinct answers but a larger number of leaves, i.e., there is some redundancy in the leaves. One way to build this new task is to increase the degree $m$ so that each step is represented multiple times, e.g., $m \to 2m$, with every edge having an equivalent "twin". This kind of redundancy does not provide any improvements to a CoT-based predictor over learning on the original tree because the number of *distinct* options at each step is still only $m$.

**An alternative strategy is to increase the depth of the reasoning tree, i.e., "thinking".** Consider extending the tree depth from $n$ to $rn$ by an expansion factor $r > 1$ while keeping the degree $m$ constant. This generates $m^{rn}$ total paths in the tree for a problem with only $m^n = N$ possible answers, creating redundancy. Fig. 6 shows an example of such a redundant tree, corresponding to the one in Fig. 5. We can define an effective number of distinct options at each step, i.e., an "effective degree" $m_{\text{eff}}$ by setting

$$N = m^n = m_{\text{eff}}^{rn} \Rightarrow m_{\text{eff}} = m^{1/r}.$$

The error of such a thinking tree is

$$\bar{E}_{\text{think}} \propto \sum_{k=1}^{rn} m_{\text{eff}}^{2/d} = (rn) m_{\text{eff}}^{2/d} = (rn) m^{2/(rd)}. \quad (10)$$

Let us define

$$\text{Thinking Gain} = \bar{E}_{\text{reason}} - \bar{E}_{\text{think}}$$

where $\bar{E}_{\text{reason}}$ was defined in Eq. (6). We visualize the theoretical difference in error with and without thinking for

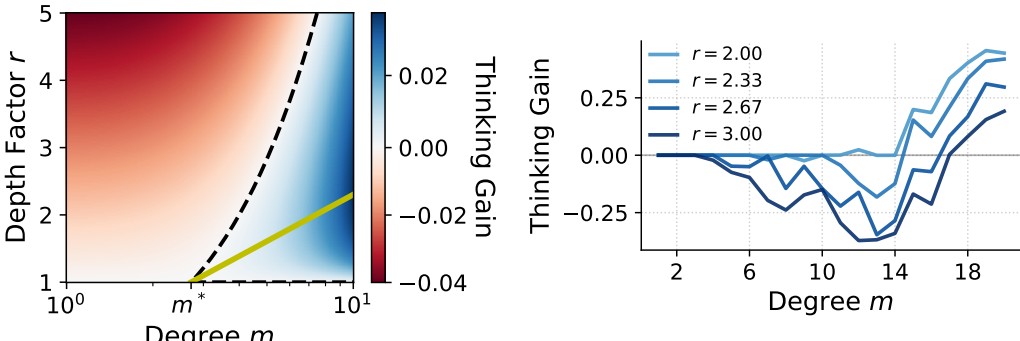

**Figure 7. Thinking, i.e., using a larger depth than necessary, is detrimental when the degree $m$ is below a threshold $m^*$ and beneficial otherwise. Left:** Heatmap values are proportional to the theoretical accuracy gain from thinking with a depth factor $r > 1$ compared to using $r = 1$ (CoT without redundancy in the tree). We use a dimension of $d = 2$ and set $cD^{-1/d} = 0.01$ for the visualization. Non-integer values of $m$ are represented for completeness. The black dashed curve indicates when the thinking gain is zero. The green line indicates the optimal depth factor $r$ such that $m_{\text{eff}} = m^{1/r} = m^*$. **Right:** The increase in accuracy from thinking for a transformer trained to solve synthetic reasoning tasks of depth $n = 3$. Across different depth factors $r$, thinking is detrimental for smaller degrees and becomes beneficial as the degree increases. Tasks with redundant trees (more leaves than the number of distinct answers) were created using a random many-to-one assignment of paths in the expanded tree. Details are provided in Section A.4.

different degrees $m$ and depth factors $r$ in Fig. 7 (left). It shows that increasing depth is harmful when the original degree $m$ is below its optimal value $m^*$ but can be beneficial when $m > m^*$ up to a depth $r$ indicated by the dashed black curve (where thinking gain is zero).

The optimal thinking gain occurs when $m_{\text{eff}} = m^* = e^{d/2}$ (green line in Fig. 7 (left)). The depth factor $r = (2/d) \ln m$ brings the effective degree $m_{\text{eff}}$ to its optimal value $m^* = e^{d/2}$ and leads to a reasoning depth

$$n^* = rn = (2/d) \ln N. \tag{11}$$

Therefore, for any sufficiently large degree of the reasoning tree, $m > m^*$, our theory predicts a single optimal depth of reasoning. Fig. 7 (right) validates this claim for a two layer transformer trained on a synthetic reasoning task. It demonstrates that thinking is detrimental for small degrees and becomes beneficial for larger degrees. The crossover points where thinking becomes beneficial increases with the depth factor $r$, matching the theoretical prediction.

The existence of an optimal depth is consistent with the experimental results in Fig. 8 on synthetic data as well as on GSM8k, MATH-500 and AIME datasets using the LLMs Qwen2.5-7B-Instruct and Deepseek-V3. We varied the reasoning length of these LLMs by using different prompts for each evaluation of the dataset. The resulting error is a convex and non-monotonic function of the number of tokens used for reasoning. In other words, reasoning for too long increases the error as our analysis above predicts.

**Remark 5 (Is there an optimal reasoning depth for LLMs?).** The experiments on real data with Qwen and Deepseek models in Fig. 8 do not definitively prove the existence of an optimal reasoning length. But it does corroborate existing results that reasoning for a larger number of

tokens can increase error (Shojaee* et al., 2025; Liu et al., 2025b). Furthermore, Kapoor et al. (2025) and Wu et al. (2026) show that the error is a convex and non-monotonic function of the reasoning length across model sizes (up to billions of parameters) and training paradigms (reinforcement learning and supervised fine-tuning). Based on our theoretical predictions as well as their empirical support (in our experiments and those from the literature), we believe that increasing reasoning length (also known as "test time scaling") does not improve accuracy arbitrarily. Instead, there is an upper bound on accuracy that is achieved at an intermediate value of reasoning depth.

**Remark 6 (Optimal CoT length decreases with LLM size and capability).** Notice that, due to limitations of the data or the model architectural, the latent states of an LLM will not capture all the relevant directions in the input domain that are required to correctly predict the next token. Therefore, the intrinsic dimension of the latent states of the LLM, call it $d'$, will be smaller than the true dimensionality of the task $d$. As a model grows in its size and capability, its latent states capture a larger proportion of the intrinsic dimensionality of the task ($d'$ increases towards $d$), and its associated optimal depth $rn = (2/d') \ln N$ will decrease. This prediction is consistent with prior experimental results indicating that the error is minimized in larger, more capable models when they are fine-tuned on shorter and more efficient reasoning traces (Wu et al., 2026).

## 5  Discussion

We next discuss our analysis within the context of some key ideas in the literature. We also include some frequently asked questions and limitations in Section B.

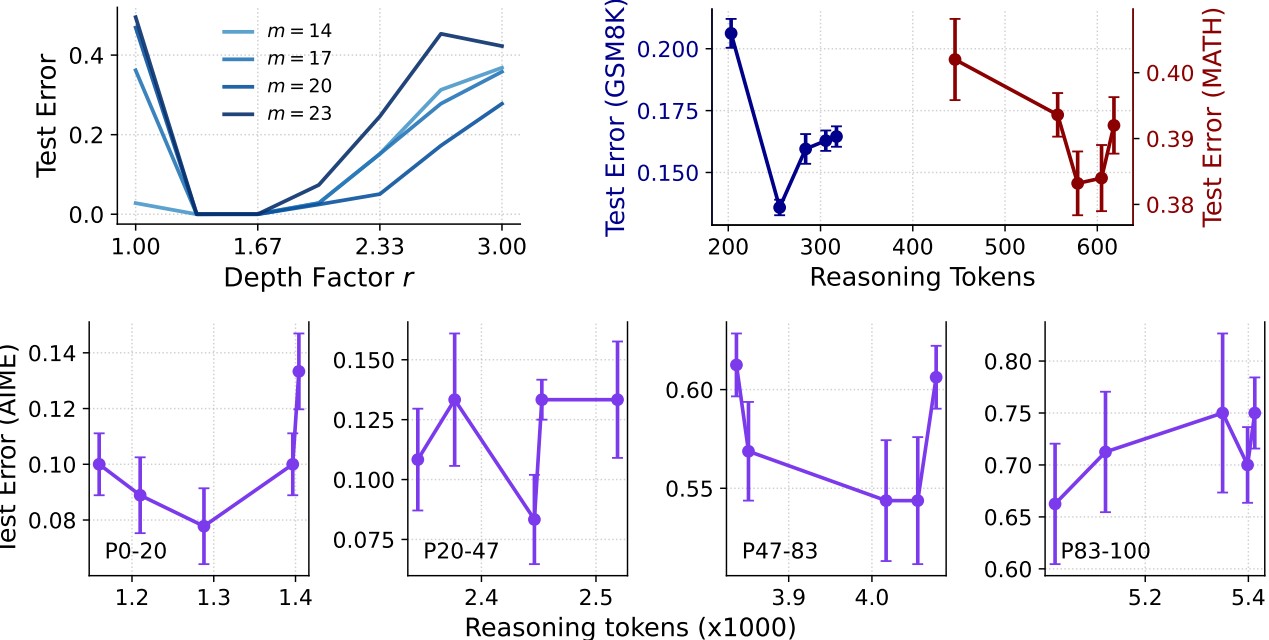

**Figure 8. Top Left:** Test error is minimized at intermediate values of the depth expansion factor $r$ across varying degrees $m$. The task is the same as in Fig. 7 (right). Details are provided in Section A.4. **Top Right and Bottom:** Test error of Qwen2.5-7B-Instruct (Yang et al., 2025b) on the GSM8k (Cobbe et al., 2021) and MATH-500 (Hendrycks et al., 2021; Lightman et al., 2023) datasets (top right) and of Deepseek-V3 (Liu et al., 2025a) on AIME problems from 2022-2024 (AIM) (bottom) is minimized at intermediate reasoning lengths. The error-bars depict the standard error of the mean across 5 replicate experiments. Instructions to the models were varied to elicit different mean reasoning lengths across the dataset. The AIME problems have variable difficulty so there is no single optimal reasoning length across the entire dataset. We therefore ordered the problems according to the mean reasoning length used to solve them (across replicates and prompts), and split them into four groups based on percentile P0-20, P20-47, P47-84 and P83-100 from left to right. This split was chosen to achieve the clearest signal. The most difficult (rightmost condition) does not show a clear optimal reasoning depth, but demonstrates the overthinking phenomenon where increasing reasoning length can degrade performance. See Section A.5 for details.

**Towards a theoretical understanding of Chain of Thought.** Recent studies have used statistical learning theory (Joshi et al., 2025) to argue that Chain of Thought leads to improved parameter-efficiency (Feng et al., 2023; Yehudai et al., 2026), and demonstrated that reasoning allows models to generalize by composing known primitives in new ways (Prystawski et al., 2023). There is also work that has used the mathematical equivalence between in-context learning and gradient descent for single-layer transformers (Von Oswald et al., 2023; Dai et al., 2023) to argue that CoT implicitly fine-tunes the weights to those with a higher likelihood of generating the correct answer.

Our work is different in two key ways. First, it relies on generic scaling laws in deep networks (Bahri et al., 2024) and is agnostic to the architecture. Second, we focus explicitly on the test error to obtain testable predictions. For instance, our framework predicts the existence of an optimal reasoning length, a phenomenon that was also observed by Wu et al. (2026) across model sizes. They explain it in terms of trade-offs between accumulated sequential error and single-step difficulty, but their model relies on specific assumptions on the scaling of error with task difficulty. In contrast, we only rely on general scaling laws. Our analysis predicts that thinking is detrimental when the reasoning tree has a small degree but beneficial when the degree is large, a prediction which we explicitly validate using synthetic reasoning tasks.

**When is test-time scaling effective?** Numerous strategies have been put forth to increase the amount of compute at test-time to improve accuracy. These include generating parallel reasoning traces (Wang et al., 2023; Snell et al., 2024), making the model self-reflect or check its work (Shinn et al., 2023; Muennighoff et al., 2025), navigating a tree of potential completions (Yao et al., 2023) and other more sophisticated strategies (Besta et al., 2024; Cheng et al., 2026). These approaches are often specific to particular problem classes (e.g., math problems, planning, etc.) and a broader understanding of these strategy is missing. Our results suggest that error is minimized when the reasoning tree has an equal degree across levels. By prioritizing important reasoning steps with these non-trivial degrees, it is possible to improve the efficiency of CoT (Li et al., 2026; 2025; Wang et al., 2026).

It has been observed that increasing reasoning length arbitrarily can cause a decline in accuracy (Shojaee* et al.,

2025; Liu et al., 2025b; Sui et al., 2025; Kapoor et al., 2025). Our results help explain when and why this "overthinking" phenomenon occurs.

**Do LLMs reason like humans?** Deductive reasoning in humans has been of interest for the better part of a century (Woodworth & Sells, 1935). In the artificial intelligence literature, deductive reasoners have been studied using formal logic (Rips, 1994; Braine & O'Brien, 1998), with recent focus on probabilistic and data-driven methods (Sejnowski, 2018). The success of CoT in LLMs challenges the dichotomy between these two paradigms. LLMs are trained to predict the next token, and yet they demonstrate what appears to be deductive reasoning, in spite of the fact that they lack explicit modules to execute intermediate logical operations (Wei et al., 2022). Our results explain how generating an intermediate chain of thought before predicting the answer, even without a direct symbolic execution of the intermediate steps, can result in a higher likelihood of arriving at the correct answer.

It is perhaps natural, then, to ask whether human reasoning implements a similar process (Yax et al., 2024). Our work can offer a way to ask targeted questions in this area. Using synthetic reasoning tasks on human subjects, we can isolate how the degree of the reasoning tree or the depth of learned reasoning traces impact their error.

**Reasoning on open-ended or creative tasks.** Our analysis relies on the assumption that the transition dynamics between reasoning steps are easy to learn, i.e., the model can accurately identify plausible next steps given its previous chain of thought. The source of error we characterize in this paper pertains to choosing one among these plausible steps. Tasks where the transition dynamics are easy to learn are known as convergent thinking tasks, where a large space of possible inputs (prompts) maps to a constrained set of valid answers (leaves) (Cropley, 2006; Guilford, 1967). Logical deduction, mathematics and code generation are examples of convergent thinking tasks. In these domains, the difficulty lies in navigating an underlying reasoning tree given a complex input. Our theory characterizes this source of error, associated with selecting the correct path within the reasoning tree, rather than the error of learning the tree itself. Our analysis does not hold for situations where the model fails to identify which next tokens are plausible. Thus, our framework may not fully capture divergent tasks, such as creative writing, where the space of acceptable answers is as large as the input space. In such regimes, the underlying tree structure, if it even exists, is often ambiguous, and extending our analysis to these open-ended domains remains a promising direction for future work.

## Acknowledgement

This work was supported by funds provided by the National Science Foundation (IS-2145164, CCF-2212519) and the National Science Foundation and DoD OUSD (R&E) under Cooperative Agreement PHY-2229929 (The NSF AI Institute for Artificial and Natural Intelligence). AN was supported by the National Science Foundation Graduate Research Fellowship Program.

## Impact Statement

This paper presents work whose goal is to advance the field of machine learning. There are many potential societal consequences of our work, none of which we feel must be specifically highlighted here.

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

# A  Details of Experiments

## A.1  Power law scaling of classification error

Fig. 1 (left) shows that test error increases as a power law with respect to the number of classes for image classification tasks based on the CIFAR-100 dataset. To demonstrate this scaling, we varied the number of classes in the task using two different methods: semantic grouping and random grouping. The semantic method merged or split Krizhevsky's original twenty superclasses (Krizhevsky & Hinton, 2009) to create $m$ balanced classes. The merging and splitting was done without any further consideration of semantic structure. The random method merged the original 100 classes to create balanced superclasses at random, without any adherence to semantic structures.

To solve these classification tasks, we fine-tuned a Vision Transformer model (`vit_tiny_patch16_224`) from the `timm` PyTorch library (Steiner et al., 2022; Dosovitskiy et al., 2021; Wightman, 2019) separately on each task. The model was pre-trained on ImageNet-21k (Ridnik et al., 2021) and fine-tuned for 25,000 iterations with a batch size of 64. The optimizer was AdamW with a learning rate of $3 \cdot 10^{-4}$, default $\beta = (0.9, 0.999)$ and a default weight decay of $0.01$. We used a cosine annealing scheduler. Test error was computed using a held-out test dataset of 10,000 images.

Fig. 1 (right) shows a similar power-law scaling on a synthetic classification task in a student-teacher setting. The input is a vector of dimension $d$ that is sampled uniformly at random from the unit sphere. The output indicates one of $m$ classes. Each class is associated with a specific unit vector (prototype), chosen uniformly at random during initialization. The ground truth label is determined by the prototype that has the maximal dot product (alignment) with the input vector. This setup corresponds to a teacher model based on a Voronoi tessellation of the sphere defined by $m$ random prototypes.

We trained a multi-layer perceptron with two hidden layers of size 128 and a ReLU activation function to solve this problem. Training was done for 500 iterations with a batch size of 512 using the AdamW optimizer with default parameters, i.e., a learning rate of $10^{-3}$, $\beta = (0.9, 0.999)$ and a weight decay of $0.01$. Testing was done using 2000 held-out samples.

## A.2  Intrinsic dimensionality is constant with respect to the reasoning step

Fig. 3 (left) shows this result using principal components analysis of the latent space of Qwen3-32B (Yang et al., 2025a) when evaluated on GSM8k (Cobbe et al., 2021) and WikiText-2 (Merity et al., 2016). We repeat this experiment in Fig. 9 with other, non-linear, dimensionality reduction methods to show the generality of the result.

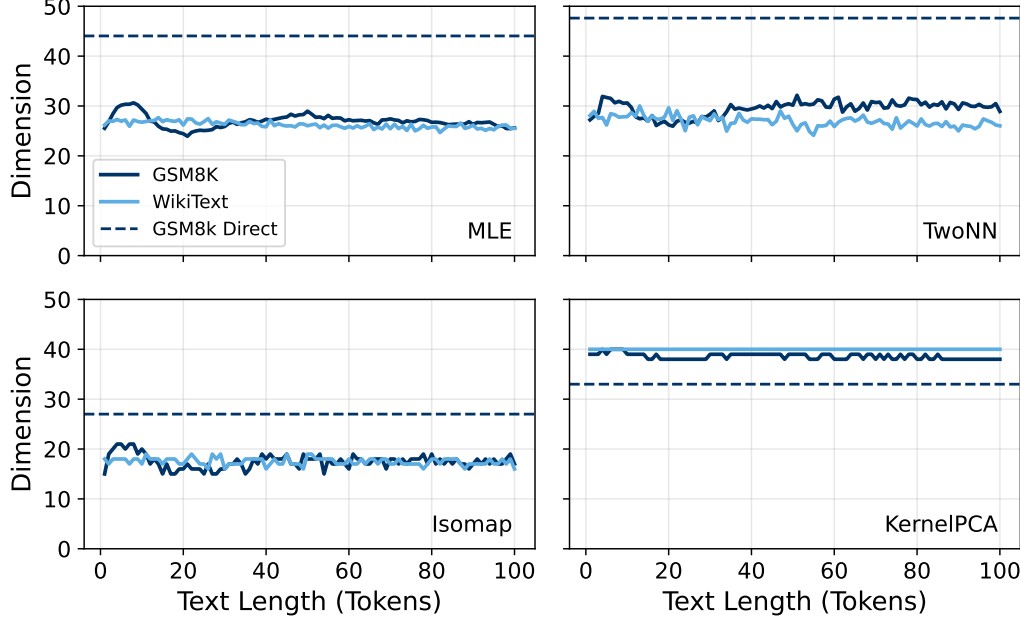

**Figure 9.** A repeat of the experiment in Fig. 3 (left) using other dimensionality reduction methods shows constant intrinsic dimension with respect to input length. We used maximum likelihood estimate (Levina & Bickel, 2004), two-nearest-neighbors (Facco et al., 2017), Isomap (Tenenbaum et al., 2000), and Kernel PCA with a radial-basis-function kernel in order from the top left, top right, bottom left and bottom right panels of the figure.

### A.3 Error decays with structure in reasoning traces

Fig. 5 (bottom) presents empirical results demonstrating that prediction error decreases as reasoning traces become more structured, i.e., they possess a similar degree at each level. These results were obtained by training Transformer models on a synthetic logical deduction task where the structure of the reasoning process was explicitly controlled.

**Consider a prototypical reasoning problem**: deduce the truth value of a "goal" variable from an input prompt or context. Such logical deductions can be made using a sequence of reasoning steps, where the value of a non-goal variable is first inferred, from which the goal variable can be deduced. The inference can alternatively be made by directly predicting the value of the goal variable with no intermediate steps. Our synthetic reasoning task was designed to model both of these cases.

The context for the task is represented by a random vector $v$ of dimension $10$. Each component of the context vector is sampled uniformly at random from the range $[-1, 1]$. The task is to determine the boolean value of a specific target node (a leaf) in a pre-defined tree structure. When this task is solved with CoT-based reasoning, the model must first identify the boolean value of the relevant node for in the first layer of the reasoning tree. These values are determined by the alignment of the context vector $v$ with a fixed reference vector $w$ of the same dimensionality. Specifically, any node $x_j$ in the first layer is equal to either $\mathbf{1}\{v \cdot w > 0\}$ or its negation. The values of subsequent nodes in the tree are determined using logical operations $f_{i \to j} \in \{\text{IDENTITY}, \text{NOT}\}$ that are assigned to each edge of the reasoning tree. The value of a child node $x_j$ is determined by applying a logical operation to its parent $x_i$, so $x_j = f_{i \to j}(x_i)$ (see Fig. 10).

This task design creates learnable patterns where a continuous set of inputs $v$ can be mapped to the same variable instantiations, but with different deductions being requested. Due this structure, this problem can be learned associatively (direct prediction). It can also be solved by reasoning based on the underlying logical structure of the problem.

**We employed a small GPT2 architecture for all experiments.** The model was instantiated using the `GPT2LMHeadModel` class from the transformers library (Radford et al., 2019). The model configuration consisted of an embedding dimension of 64, 2 Transformer layers and 4 attention heads. Each task was instantiated with a fixed degree of $m = 3$ and varying depth $n$. The training dataset size was a function of the depth $n$ such that the model was trained on $15{,}000 \times n^{0.7}$ samples. Training was done for 20 epochs and with a batch size of 256 using the AdamW optimizer with a learning rate of $2.5 \cdot 10^{-3}$, default $\beta$ values of $(0.9, 0.999)$ and a default weight decay of $0.01$. We also employed a linear learning rate scheduler such that the first 10% of training steps were used as a warmup, and then the learning rate decayed linearly to 0 after the initial period. Error was computed using a test set of 5,000 unseen samples and greedy decoding as the inference strategy.

During inference, the context vector $v$ is projected into the model's input embedding sequence using a learnable linear layer, i.e., $v$ is converted into the embedding for the first token position. This initial embedding is followed by text specifying the target leaf node, e.g., "Target: X5". The text is processed using a Byte-Pair Encoding (BPE) tokenizer trained on a subset of 100,000 samples with a vocabulary size of 5,000 tokens. Each point in Fig. 5 (bottom) represents the mean of 10 replicate experiments such that the identity and negation operations on edges of the tree, the training set and the test set were varied across replicates. In that figure, a structure of $k/n$ corresponds to a degree of $3$ for the first $k - 1$ layers, then a degree of $3^{n-k+1}$ for the $k$'th layer, and then a trivial branching factor of $1$ until the final $n$'th layer. In Fig. 5 (bottom), tasks with a

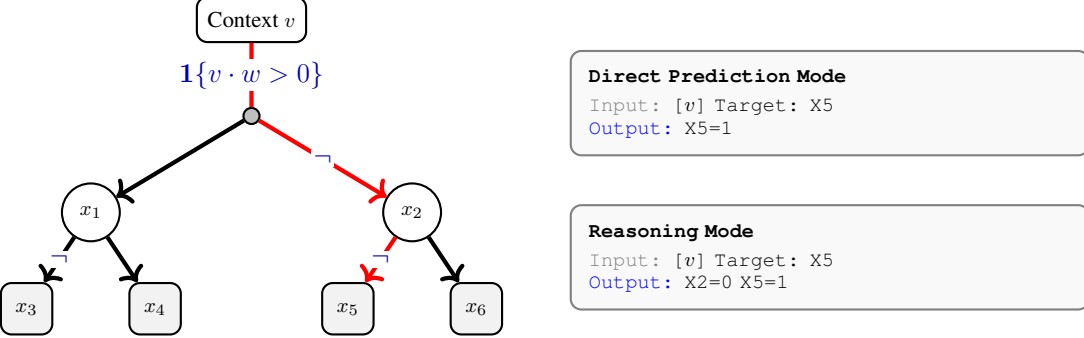

**Figure 10. A diagram of the synthetic logical deduction task. Left:** The boolean value of the root (shaded dot) is determined by the dot product of a random vector $v$ representing the context, and a fixed reference vector $w$. The value propagates down the tree according to fixed logical operations (IDENTITY or NOT $\neg$). For example, if $\mathbf{1}\{v \cdot w > 0\} = 1$, then $x_2 = 0$ and $x_5 = \neg x_2 = 1$. **Right:** Comparison of training targets for reasoning and non-reasoning models. The "Direct" model predicts the final leaf value immediately. The "Reasoning" model must predict the state of every node along the highlighted red path in the tree before predicting the final leaf value.

structure of $k = 1$ were often learned with a lower error compared to direct prediction. This is odd, since in both cases, the model has to distinguish between $3^n$ branches in its first layer. We think this occurs because each token in the model output is not equivalent to a single reasoning step, i.e., a reasoning step such as "X1843=0", might require the model to generate multiple tokens to represent. Therefore, there is likely some implicit structure in the model output, even when $k = 1$, that is a result of how the tokenizer decomposes each reasoning step.

### A.4 CoT with thinking

Fig. 7 (right) and Fig. 8 (top left) present empirical results demonstrating that thinking, i.e., increasing the depth of the reasoning tree to include redundant paths, is most beneficial when the degree $m$ is large and the depth factor $r$ takes an intermediate value. These results were obtained by training Transformer models on a synthetic reasoning task similar to the one in Fig. 10, but with increased tree depth.

Recall the task in Fig. 10. The input prompt consists of a context vector $v$ and a target leaf node, e.g., $x_5$. The boolean value of the target node can be deduced by first computing the product $\mathbf{1}\{v \cdot w > 0\}$ and then applying a sequence of logical operations defined by the tree structure.

To model thinking, we first define a task using the same procedure as the previous section. Then, we construct an augmented tree with increased depth $rn$ for depth factor $r > 1$, while maintaining the original degree $m$. To avoid confusion, nodes in the augmented tree are labeled with "Y" while those in the original tree are labeled with "X". The augmented tree has roughly $m^{rn}/m^n$ redundant leaves for every leaf in the original reasoning tree. Thus, each leaf $x_k$ from the original tree is assigned, at random, to approximately $m^{(r-1)n}$ leaves in the augmented tree. The boolean operations (IDENTITY or NOT) on the edges learning up final layer are constrained to ensure that the value of every leaf in the augmented tree is equal to the value of the corresponding leaf $x_k$ in the original tree.

Fig. 11 illustrates this setup. The model is prompted with the target node from the original tree (e.g., "Target: X5"). It is trained to predict a deeper path through the thinking tree to retrieve the value of this target node. It has multiple reasoning paths it can choose which will give it the same correct answer, as opposed to in the original tree in Fig. 10 where there is only one path per target. The training samples represent each reasoning path with equal likelihood.

Apart from changing the training set size to a fixed value of 50,000 samples, we employed the exact same model architecture, training hyper-parameters and testing procedure as described in the previous section to generate Fig. 7 (right) and Fig. 8 (top left). Each point in the figure represents the mean of 20 replicate experiments such that the identity and negation operations on edges of the tree, the training set and the test set were varied at random across replicates.

### A.5 Accuracy on GSM8k, Math-500 and AIME against reasoning length

Fig. 8 (top right) shows that error on the GSM8k test set (Cobbe et al., 2021) and on MATH-500 (Hendrycks et al., 2021; Lightman et al., 2023) is minimized when prompting Qwen2.5-7B-Instruct (Yang et al., 2025b) to generate answers with an intermediate reasoning length. Fig. 8 (bottom) shows the same result when evaluating Deepseek-V3 (Liu et al., 2025a) on

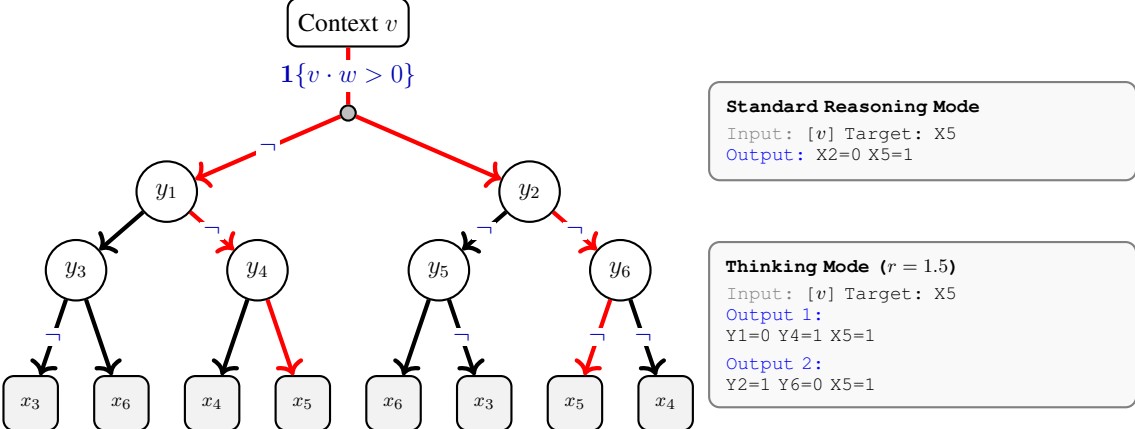

**Figure 11.** Diagram of an augmented thinking tree corresponding to the original tree from Fig. 10. **Left:** The tree depth is increased from $n = 2$ to $rn = 3$, creating redundant leaves. The edge operations, identity and negation ($\neg$) are chosen so that the values of the leaf nodes are consistent with their counterparts on the original tree. The paths to the two leaves corresponding to $x_5$ are highlighted in red. **Right:** Comparison of training targets. The thinking model can generate multiple valid reasoning traces to deduce the same final answer.

AIME problems (AIM). During evaluation of the Qwen model, we used top-$p$ decoding where $p = 0.9$ and a temperature of $1.0$. The Deepseek model was evaluated through using standard stochastic sampling with a temperature of $1.0$. The error and reasoning length for each question was averaged over $5$ replicate answer generations. These results were than averaged over the entire dataset to form a single point on the figure. Each set of evaluations on the dataset was done using a different set of instructions to the model that was provided as a system prompt before the question text. Each instruction was designed to elicit varied mean response lengths during evaluation. We have listed the instructions for the GSM8k dataset below in the order of their elicited mean response length.

> 1. You are a helpful assistant. Solve the math problem. Show your work. Only show important steps. Prepend #### to your final answer.
> 2. You are a helpful assistant. Solve the math problem. Show your work step by step. Prepend #### to your final answer.
> 3. You are a helpful assistant. Solve the math problem. Show your work step by step. Check each step to make sure it is correct. Prepend #### to your final answer.
> 4. You are a helpful assistant. Solve the math problem. Show your work step by step. Explain each step. Explicitly check each step to make sure it is correct. Prepend #### to your final answer.
> 5. You are a helpful assistant. Solve the math problem. Show your work step by step. Explain each step. Explicitly check each step to make sure it is correct. Then, check each step again to make sure it is correct. Prepend #### to your final answer.

Each instruction in this list builds on the previous one. The first level encouraged the model to skip steps in its reasoning, while the second asked it to show all its steps. Following that, future instructions asked the model to explicitly check the result of, and explain the logic behind each step. By prompting the model in this way, we were able to generate responses of varying length, and where each response made sense in the context of the problem. We also provided few-shot examples to the model for each of the conditions above. The corresponding examples for each instruction are listed below.

> 1. Natalia sold clips to 48 of her friends in April, and then she sold half as many clips in May. How many clips did Natalia sell altogether in April and May? \n Natalia sold 48/2=24 clips in May. She sold 72 clips altogether in April and May. #### 72 \n Weng earns $12 an hour for babysitting. Yesterday, she just did 50 minutes of babysitting. How much did she earn? \n Weng earns 12/60=1/5 dollars per minute. She earns 10 dollars in 50 minutes. #### 10 \n
>
> 2. Natalia sold clips to 48 of her friends in April, and then she sold half as many clips in May. How many clips did Natalia sell altogether in April and May? \n Natalia sold 48/2=24 clips in May. She sold 48 clips in April. She sold 48+24 = 72 clips altogether in April and May. #### 72 \n Weng earns $12 an hour for babysitting. Yesterday, she just did 50 minutes of babysitting. How much did she earn? \n Weng earns 12/60 = 1/5 dollars per minute. In 50 minutes, she earns 1/5 dollars per minute*50 minutes=10 dollars. #### 10 \n
>
> 3. Natalia sold clips to 48 of her friends in April, and then she sold half as many clips in May. How many clips did Natalia sell altogether in April and May? \n Natalia sold 48/2=24 clips in May. Checking, half of 48 is 24. She sold 48 clips in April. She sold 48+24 = 72 clips altogether in April and May. Checking, 48+24=72 is correct. #### 72 \n Weng earns $12 an hour for babysitting. Yesterday, she just did 50 minutes of babysitting. How much did she earn?\n Weng earns 12/60 = 1/5 dollars per minute. Checking, 12/60=1/5 because 60/12=5. In 50 minutes, she earns 1/5 dollars per minute*50 minutes=10 dollars. Checking, we must multiply time working with the earning rate, 50 * 1/5=10 is correct. #### 10 \n
>
> 4. Natalia sold clips to 48 of her friends in April, and then she sold half as many clips in May. How many clips did Natalia sell altogether in April and May? \n I should calculate how many clips Natalia sold in May. Natalia sold 48/2=24 clips in May. Checking, half of 48 is 24. She sold 48 clips in April. The total will be the sum of the clips sold in April and May. She sold 48+24 = 72 clips altogether in April and May. Checking, 48+24=72 is correct. #### 72 \n Weng earns $12 an hour for babysitting. Yesterday, she just did 50 minutes of babysitting. How much did she earn? \n I should calculate Weng's earning rate per minute. Weng earns 12/60 = 1/5 dollars per minute. Checking, 12/60=1/5 because 60/12=5. To calculate the money earned, I must multiply the earning rate with the time spent working. In 50 minutes, she earns 1/5 dollars per minute*50 minutes=10 dollars. Checking, I must multiply time working with the earning rate, 50 * 1/5=10 is correct. #### 10 \n
>
> 5. Natalia sold clips to 48 of her friends in April, and then she sold half as many clips in May. How many clips did Natalia sell altogether in April and May? \n I should calculate how many clips Natalia sold in May. Natalia sold 48/2=24 clips in May. Checking, half of 48 is 24. Checking again, 48 divided by 2 is 24. She sold 48 clips in April. The total will be the sum of the clips sold in April and May. She sold 48+24 = 72 clips altogether in April and May. Checking, 48+24=72 is correct. Checking again, the sum of 48 and 24 is 72. #### 72 \n Weng earns $12 an hour for babysitting. Yesterday, she just did 50 minutes of babysitting. How much did she earn? \n I should calculate Weng's earning rate per minute. Weng earns 12/60 = 1/5 dollars per minute. Checking, 12/60=1/5 because 60/12=5. Checking again, 12/60=0.2 which is equal to 1/5. To calculate the money earned, I must multiply the earning rate with the time spent working. In 50 minutes, she earns 1/5 dollars per minute*50

minutes=10 dollars. Checking, I must multiply time working with the earning rate, 50*1/5=10 is correct. Checking again, 50*1/5 is 10. #### 10 \n

The Math-500 and AIME datasets were evaluated with the same prompts as each other and similar to those of GSM8k (in this case we boxed the answer). The evaluations for both these datasets used the same few-shot examples that were different than those used for GSM8k. The prompts are listed below in the order of their elicited mean response length.

1. You are a helpful assistant. Solve the math problem. Show your work. Only show important steps. Output the final answer in a box.
2. You are a helpful assistant. Solve the math problem. Show your work step by step. Output the final answer in a box.
3. You are a helpful assistant. Solve the math problem. Show your work step by step. Check each step to make sure it is correct. Output the final answer in a box.
4. You are a helpful assistant. Solve the math problem. Show your work step by step. Explain each step. Explicitly check each step to make sure it is correct. Output the final answer in a box.
5. You are a helpful assistant. Solve the math problem. Show your work step by step. Explain each step. Explicitly double-check each step to make sure it is correct. Output the final answer in a box.

And the few shot examples for these two datasets are listed below in the same order.

1. Let $\[f(x) = \left\{ \begin{array}{cl} ax+3, &\text{ if }x>2, \\ x-5 &\text{ if } -2 \le x \le 2, \\ 2x-b &\text{ if } x <-2. \end{array} \right.\]$Find $a+b$ if the piecewise function is continuous (which means that its graph can be drawn without lifting your pencil from the paper). \n For the piecewise function to be continuous, the cases must meet at $2$ and $-2$. This means $ax+3$ must equal $x-5$ when $x=2$ and, therefore, $a=-3$. Similarly, $x-5$ must equal $2x-b$ at $x=-2$, which implies $b=3$. Then, the sum $a+b=0$. \boxed{0}$ \n What is the value of $9^3 + 3(9^2) + 3(9) + 1$? \n This expression is a cubic polynomial with coefficients $(1,3,3,1)$ in decreasing order of degree. This means the expression is equal to $(9+1)^3$. Thus, its value is $10^3$. \boxed{1000}$

2. Let $\[f(x) = \left\{ \begin{array}{cl} ax+3, &\text{ if }x>2, \\ x-5 &\text{ if } -2 \le x \le 2, \\ 2x-b &\text{ if } x <-2. \end{array} \right.\]$Find $a+b$ if the piecewise function is continuous (which means that its graph can be drawn without lifting your pencil from the paper). \n For the piecewise function to be continuous, the cases must meet at $2$ and $-2$. This means $ax+3$ must equal $x-5$ when $x=2$. Therefore, $2a+3=2-5$ and $2a+3=-3\rightarrow$ $2a=-6$ so $a=-3$. Similarly, $x-5$ must equal $2x-b$ at $x=-2$. This means $2(-2)-b=-2-5$ so $-4-b=-7\rightarrow -b=-3$ so $b=3$. Then, the sum $a+b=0$. \boxed{0}$ \n What is the value of $9^3 + 3(9^2) + 3(9) + 1$ \n This expression is a cubic polynomial with coefficients $(1,3,3,1)$ in decreasing order of degree. The polynomial $(x+1)^3=x^3+3x^2+3x+1$ has the same coefficients. This means the expression is equal to $(x+1)^3$ for $x=9$ which is $(9+1)^3$. Thus, its value is $10^3$. \boxed{1000}$

3. Let $\[f(x) = \left\{ \begin{array}{cl} ax+3, &\text{ if }x>2, \\ x-5 &\text{ if } -2 \le x \le 2, \\ 2x-b &\text{ if } x <-2. \end{array} \right.\]$Find $a+b$ if the piecewise function is continuous (which means that its graph can be drawn without lifting your pencil from the paper). \n For the piecewise function to be continuous, the cases must meet at $2$ and $-2$. This means $ax+3$ must equal $x-5$ when $x=2$. Therefore, $2a+3=2-5$ and $2a+3=-3\rightarrow$ $2a=-6$ so $a=-3$. Checking by substitution, $2a+3=2-5$ means $2(-3)+3=-3\rightarrow -6+3=-3$ which is a true expression. Similarly, $x-5$ must equal $2x-b$ at $x=-2$. This means $2(-2)-b=-2-5$ so $-4-b=-7\rightarrow -b=-3$ so $b=3$. Checking by substitution, $2(-2)-3=-2-5\rightarrow -4-3=-7$ which is a true expression. Then, the sum $a+b=0$. \boxed{0}$ \n What is the value of $9^3 + 3(9^2) + 3(9) + 1$? \n This expression is a cubic polynomial with coefficients $(1,3,3,1)$ in decreasing order of degree. The polynomial $(x+1)^3=x^3+3x^2+3x+1$ has the same coefficients. Checking, $(x+1)^3=(x+1)(x+1)(x+1)=(x+1)(x^2+2x+1)=x^3+2x^2+x+x^2+2x+1=x^3+3x^2+3x+1$ which yields the correct coefficients. This means the expression is equal to $(x+1)^3$ for $x=9$ which is $(9+1)^3$. Thus, its value is $10^3$. \boxed{1000}$

4. Let $\[f(x) = \left\{ \begin{array}{cl} ax+3, &\text{ if }x>2, \\ x-5 &\text{ if } -2 \le x \le 2, \\ 2x-b &\text{ if } x <-2. \end{array} \right.\]$Find $a+b$ if the piecewise function is continuous (which means that its graph can be drawn without lifting your pencil from the paper). \n For the piecewise function to be continuous, the cases must meet at $2$ and $-2$. This means $ax+3$ must equal $x-5$ when $x=2$. This is because the piecewise function is equal to $ax+3$ above $x=2$ and is equal to $x-5$ below $x=2$. Both lines must meet at exactly $x=2$ for the function to be continuous. Therefore, $2a+3=2-5$ and $2a+3=-3\rightarrow 2a=-6$ so $a=-3$. Checking by substitution, $2a+3=2-5$ means $2(-3)+3=-3\rightarrow -6+3=-3$ which is a true expression. Similarly, $x-5$ must equal $2x-b$ at $x=-2$. This is because the piecewise function is equal to $x-5$ for $x$ immidiately larger than $-2$, and the function is equal to $2x-b$ when $x$ is smaller than $-2$. Both lines must meet at $x=-2$. This means $2(-2)-b=-2-5$ so $-4-b=-7\rightarrow -b=-3$ so $b=3$. Checking by substitution, $2(-2)-3=-2-5\rightarrow -4-3=-7$ which is a true expression. Then, the sum $a+b=0$. \boxed{0}$ \n What is the value

of $9^3 + 3(9^2) + 3(9) + 1$?\n This expression is a cubic polynomial with coefficients $(1,3,3,1)$ in decreasing order of degree. Identifying the polynomial nature in the expression will help us solve it. Given that the expression is a polynomial, we can try to factorize it. The polynomial $(x+1)^3=x^3+3x^2+3x+1$ has the same coefficients as the expression. Checking, $(x+1)^3=(x+1)(x+1)(x+1)=(x+1)(x^2+2x+1)=x^3+2x^2+x+x^2+2x+1=x^3+3x^2+3x+1$ which yields the correct coefficients. This means the expression is equal to $(x+1)^3$ for $x=9$ which is $(9+1)^3$. By writing the expression in this factorized form, we can evaluate the entire expression by computing the simplified sum $9+1=10$ within the parentheses. Thus, its value is $10^3$. \boxed{1000}$

5. Let \[f(x) = \left\{ \begin{array}{cl} ax+3, &\text{ if }x>2, \\ x-5 &\text{ if } -2 \le x \le 2, \\ 2x-b &\text{ if } x <-2. \end{array} \right.\]Find $a+b$ if the piecewise function is continuous (which means that its graph can be drawn without lifting your pencil from the paper). \n For the piecewise function to be continuous, the cases must meet at $2$ and $-2$. This means $ax+3$ must equal $x-5$ when $x=2$. This is because the piecewise function is equal to $ax+3$ above $x=2$ and is equal to $x-5$ below $x=2$. Both lines must meet at exactly $x=2$ for the function to be continuous. Therefore, $2a+3=2-5$ and $2a+3=-3\rightarrow 2a=-6$ so $a=-3$. Checking by substitution, $2a+3=2-5$ means $2(-3)+3=-3\rightarrow -6+3=-3$ which is a true expression. Checking again, $2a+3=2-5\rightarrow 2a+3=-3\rightarrow 2a=-6$ so $a=-3$. Similarly, $x-5$ must equal $2x-b$ at $x=-2$. This is because the piecewise function is equal to $x-5$ for $x$ immidiately larger than $-2$, and the function is equal to $2x-b$ when $x$ is smaller than $-2$. Both lines must meet at $x=-2$. This means $2(-2)-b=-2-5$ so $-4-b=-7\rightarrow -b=-3$ so $b=3$. Checking by substitution, $2(-2)-3=-2-5\rightarrow -4-3=-7$ which is a true expression. Checking again, $2(-2)-b=-2-5\rightarrow -4-b=-7\rightarrow -b=-3$ so $b=3$. Then, the sum $a+b=0$. \boxed{0}$ \n What is the value of $9^3 + 3(9^2) + 3(9) + 1$? \n This expression is a cubic polynomial with coefficients $(1,3,3,1)$ in decreasing order of degree. Identifying the polynomial nature in the expression will help us solve it. Given that the expression is a polynomial, we can try to factorize it. The polynomial $(x+1)^3=x^3+3x^2+3x+1$ has the same coefficients as the expression. Checking, $(x+1)^3=(x+1)(x+1)(x+1)=(x+1)(x^2+2x+1)=x^3+2x^2+x+x^2+2x+1=x^3+3x^2+3x+1$ which yields the correct coefficients. Checking again, $(x+1)^3$ can be expanded to be $(x+1)(x+1)(x+1)=(x^2+2x+1)(x+1)=x^3+2x^2+x+x^2+2x+1=x^3+3x^2+3x+1$, matching the coefficients in the expression. This means the expression is equal to $(x+1)^3$ for $x=9$ which is $(9+1)^3$. By writing the expression in this factorized form, we can evaluate the entire expression by computing the simplified sum $9+1=10$ within the parentheses. Thus, its value is $10^3$. \boxed{1000}$

# B   Frequently Asked Questions (FAQs)

1. **Are there any practical guidelines for building real-world reasoning systems?**

   A theoretical understanding of CoT-based reasoning can help steer future research towards ideas that might have a greater chance of being successful.

   A potentially interesting consequence of our analysis is that it implies that reasoning traces do not have to be human-understandable. As long as the next-token predictions have a tree-like structure researchers can have the same improvements in accuracy on top of CoT-based predictions. Thus, when curating training data, it is not necessary to always use expensive human-generated reasoning traces in natural language. In fact, it has been noticed that human-readable reasoning traces can be less effective (Pfau et al., 2024).

   Second, our analysis helps characterize when when CoT-based reasoning is a useful strategy and how to implement it for maximal efficiency. Given a dataset, the analysis that we performed in Fig. 5 (bottom) can be conducted very easily by simply building a prefix tree (trie) on the encoded data. Our analysis shows that if the degree of such a trie is roughly the same across depth, then this data is beneficial for training CoT-based predictors. LLMs are known to "overthink" on simple tasks which results in diminished accuracy (Shojaee* et al., 2025). Even in the cases where generating a long chain of thought helps with accuracy, the process of generating that text uses a significant amount of resources. It may be possible to one day develop a model architecture or training paradigm that automatically augments its training data with reasoning traces of optimal depth and degree.

   Third, our analysis suggests that the tree-like structure in CoT can be used to improve accuracy on even standard classification tasks, not just text prediction using LLMs. Researchers have found success from chaining together a sequence of predictions in diverse domains such as protein structure analysis (Hayes et al., 2025), robotics (Antonova et al., 2023; Driess et al., 2023) and image classification (Heisele et al., 2003; Park et al., 2025).

2. **"This is not chain-of-thought", "This is not how reasoning works", "This is not how thinking is implemented in an LLM"**

   In our theory, CoT-based reasoning is characterized by a sequence of predictions made by a model on intermediate

classification sub-tasks. The set of all possible sequences of predictions (chains of thought) can be organized into a tree where the intermediate nodes represent reasoning tokens and the leaves correspond to answers. We use the term "thinking" to refer to cases where this reasoning tree has greater depth than is strictly necessary for the task, resulting in redundancy (multiple leaves corresponding to the same answer).

These definitions are precise in the context of the paper and, in broad strokes, they do capture how LLMs use CoT. At each reasoning step, a LLM predicts a probability distribution over the next token and samples from that distribution, not unlike a classifier. While there are certainly many details of real-world LLMs that are not captured by our description, e.g., different sampling strategies, architectural choices, and specific patterns in real-world data, it provides a coherent theory that matches many existing observations in the literature. This is a reasonable first-step to understanding how chain-of-thought decomposes complex tasks into a sequence of simpler ones.

3. **What is the optimal degree and depth in real problems? How can you check the degree $m$ of a given task?** Our scaling law in Eq. (4) as well as prior empirical and theoretical results (Bahri et al., 2024; Kaplan et al., 2020) indicate that error should scale as roughly $D^{-1/d}$. By evaluating the test error of a model while varying the size of the dataset $D$, one can estimate the scaling exponent and, therefore, estimate the dimension of the task $d$. The optimal degree is then $m^* = e^{d/2}$ and the optimal depth is $(2/d) \ln N$ which can both be estimated. The challenge with this approach is that the scaling of loss or error with respect to dataset size is typically measured during the pretraining phase, where no single task is isolated. Computing these quantities for a specific real-world task might be more difficult.

In a synthetic task with controllable values of the degree $m$ and the depth $n$ of the reasoning tree, it is possible to estimate the optimal degree $m^*$ using experiments that are similar to those in Fig. 7 (right) and Fig. 8 (top left). In the former figure, by computing the degrees $m$ where thinking becomes beneficial (crossover points) for different values of the depth factor $r$, one can extrapolate the optimal degree $m^*$ by tracing the dashed black line in Fig. 7 (left). In addition, similar to the latter figure, the depth factors $r$ where error is minimized for different degrees $m$ and fixed task size $N$ can be used to estimate the optimal degree $m^*$ by tracing points on the yellow line in Fig. 7 (left).

It is difficult to measure the degree in real-world data, since benchmark datasets do not capture every possible reasoning path. Calculating the optimal degree for real-world problems is even more difficult since it requires estimating the intrinsic dimensionality of the next-token prediction task. It is possible to make a comparison between the degree of the reasoning traces used to solve a task and the optimal degree for that task, as estimated by the dimensionality of the model latent states. Specifically, when thinking, i.e., reasoning on a deeper tree, improves error, we expect that the degree $m$ of the reasoning tree is larger than optimal. This is often the case in real-world tasks (Guo et al., 2025; Xu et al., 2025). On the other hand, when thinking is detrimental (Wu et al., 2026; Shojaee* et al., 2025), we expect that the degree of the reasoning tree is too small.

4. **How would this analysis change if the LLM were to use tree-of-thought, graph-of-thoughts, self-consistency etc.?**

We have assumed each reasoning step is sampled from the predicted probability distribution over the next token. However, it is common to use more sophisticated methods to improve the effectiveness of CoT-based reasoning (Yao et al., 2023; Besta et al., 2024; Wang et al., 2023). One such method is self-consistency, where a model stochastically generates several answers and picks the one that was generated most often (Wang et al., 2023). Using our setup, it is indeed possible to calculate the expected error for self-consistency, and perhaps also for ideas like tree-of-thought (Yao et al., 2023). It boils down to modeling the number of decisions $m$ at each step, the number of steps $n$ and the probability of error at each step $\bar{E}$ in Section 2. We hope to do so in future work.

5. **How might training with reinforcement learning change the results of this analysis?**

Reinforcement learning (RL) has been shown to improve reasoning capabilities beyond the limits of supervised fine-tuning on human reasoning traces (Xie et al., 2025; Chu et al., 2025). So long as the inference procedure at test time involves a chain of thought, our analysis holds, even for models that are trained using RL.

In fact, one interesting avenue for future exploration could involve investigating the hypothesis that training on reward signals implicitly encourages models to maximize the shared structure in their reasoning traces and optimize their reasoning length. Previous research has found that RL can tune reasoning length towards an optimal intermediate value in both frontier models as well as in smaller transformers (Wu et al., 2026). However, while reinforcement learning has become a key training method to elicit reasoning in LLMs (Comanici et al., 2025; OpenAI, 2024; Guo et al., 2025), it suffers from high sample complexity (Lightman et al., 2023; Yue et al., 2026). It is an open question whether there are alternative training protocols which allow a model to optimize the shared structure in and length of its reasoning traces

with high sample-efficiency. For example, it might be useful to encourage the generation of structured filler tokens before generating a next word (Pfau et al., 2024).

