# OpenReview forum: "How does Chain of Thought decompose complex tasks?"
_ICML.cc/2026/Conference — ICML 2026 regular_

### Official Review · Reviewer_4Bg9 · 2026-03-11

**Soundness:** 3
**Presentation:** 2
**Significance:** 3
**Originality:** 3
**Overall Recommendation:** 4
**Confidence:** 3

**Summary:**

This paper presents a rigorous theoretical and empirical investigation into how Chain of Thought (CoT) reasoning decomposes complex tasks for large language models (LLMs), resolving conflicting empirical findings about when CoT improves performance and when excessive "thinking" harms it. The core framework models reasoning tasks as tree structures—where the root is a prompt, leaves are ground-truth answers, and edges represent plausible next reasoning steps—and links key tree properties (branching degree at each level, reasoning depth, redundant paths) to prediction error using statistical learning theory and scaling laws.
Empirically, the work validates these theoretical predictions across synthetic logical deduction tasks (where tree structure, degree, and depth are fully controllable) and real-world benchmarks: synthetic transformer experiments confirm that more structured reasoning trees yield lower error, CIFAR-100 classification validates the power-law error scaling, and GSM8k evaluations show that LLM test error is minimized at an intermediate CoT length (not infinite reasoning).
Overall, the paper advances the theoretical understanding of CoT reasoning, provides actionable guidelines for building effective LLM reasoning systems, and opens new avenues for extending the framework to other reasoning paradigms.

**Compliance With Llm Reviewing Policy:**

Affirmed.

**Key Questions For Authors:**

1. How would your tree-structure framework apply to advanced CoT variants like Tree of Thought (ToT), Graph of Thought (GoT), or self-consistency, and what new theoretical predictions would your model generate for these methods? You briefly note that your framework could extend to advanced CoT variants but do not elaborate on this. A concrete extension of your model to one or two widely used variants would demonstrate your framework’s generalizability.

2. Have you tested your optimal reasoning depth prediction on larger, frontier LLMs, especially those like DeepSeek-R1 that are known for long reasoning paths—would be critical to validating your theory’s broad applicability.?

3. Your paper provides actionable practical guidance but does not include an end-to-end demonstration of a system built using your theory. A concrete implementation would translate your theoretical claims into a tangible improvement for practitioners.

**Limitations:**

yes

**Strengths And Weaknesses:**

Strengths:
1. The paper’s core theoretical contributions—including the power-law scaling of classification error with the number of classes and intrinsic dimensionality, and the derivation of an optimal branching degree for reasoning trees—are mathematically sound and built on well-established statistical learning theory (e.g., Lipschitz continuity, nonparametric estimation bounds). All theoretical claims are supported by explicit derivations, and simplifying assumptions are clearly stated and empirically validated.
2. The authors balance mathematical rigor with plain language explanations of key concepts. The FAQ section further demystifies abstract ideas, making the work accessible to both theoretical ML researchers and LLM practitioners.
3. For years, CoT research has been dominated by empirical work, with little theoretical grounding in statistical learning theory. This paper closes that gap by linking CoT reasoning to core classification and scaling law theory, making CoT a predictable rather than a black box method. This allows researchers to design CoT systems based on first principles, not just trial and error.
4. The paper’s core insight—modeling all plausible CoT reasoning paths as a tree with roots (prompts), leaves (answers), and edges (reasoning steps)—is a novel abstraction that unifies CoT research under a single theoretical framework.

Weaknesses:
1. Dense mathematical notation in key sections: Sections 2 (classification error scaling) and 3 (CoT error bounds) use dense notation without a dedicated glossary, which may be a barrier for readers without a strong background in statistical learning theory.
2. The paper explicitly restricts its framework to convergent thinking tasks (where a large input space maps to a constrained set of answers), which is a necessary boundary but limits its immediate impact on open-ended LLM applications.
3. The paper does not introduce a new CoT method, algorithm, or LLM architecture—its originality lies in theoretical abstraction and empirical validation, not in novel engineering. While this is a strength for a theoretical paper, it means the work does not provide a directly implementable new tool for practitioners.

---

> ### Author Rebuttal · Authors · 2026-03-31
>
> We appreciate your feedback. We are glad you think our investigation is rigorous and provides actionable guidelines for building effective LLM reasoning systems. We address your comments below.
>
> > **dedicated glossary**
>
> Great point. We will add this.
>
> > **restricts to convergent thinking tasks…**
>
> Indeed, our analysis is not applicable to tasks such as writing an essay. However, we believe that convergent thinking tasks are exactly where reasoning as a deduction strategy is necessary. Open-ended tasks like writing an essay might not require precise reasoning, e.g., the plot of a fiction novel needs to have consistent themes but the “correctness” of the text is nebulous.
>
> > **The paper does not provide a directly implementable new tool for practitioners**
>
> As mentioned in the discussion, our results indicate that arbitrarily scaling up the thinking time of an LLM does not necessarily improve accuracy.
>
> FAQ 1 in the Appendix discusses this further. An improved theoretical understanding of CoT can help steer future research towards ideas that are more likely to be successful. Our analysis also indicates that CoT does not have to be human-understandable. It is the shared structure of the reasoning traces that yields the benefit in accuracy. Thus, it should be possible to cheaply generate training data with a tree-like structure to solve reasoning problems without relying entirely on expensive human-generated reasoning traces. In fact, it may be possible to develop a training paradigm that automatically augments the training data with reasoning traces of optimal depth and degree. Finally, we expect that the same tree-like structure present in standard CoT can be used to decompose other classification tasks such as image classification, protein structure analysis, robotics, etc.
>
> > **Your paper provides actionable practical guidance but does not include an end-to-end demonstration of a system built using your theory.**
>
> Our paper does not implement a new technique for CoT. However, we do have a number of end-to-end demonstrations in the paper. The experiments in Fig 1, 5, 7, 8, 9 (left) use a GPT-style transformer trained on text data from synthetic tasks for next-token prediction (see Appendix A.2). Our experiments on mathematical reasoning datasets in Fig 9 (right) and on GSM8k and MATH-500 using Qwen, Gemma and now DeepSeek models (https://imgur.com/a/MYEFjEj) are not “end-to-end” experiments but they validate the theory.
>
> > **...Tree of Thought (ToT), Graph of Thought (GoT), or self-consistency…**
>
> So long as an LLM produces a single sequence of tokens before the answer (e.g., in backtracking or self-reflection) all our results hold. Our analysis does not hold directly for strategies such as majority voting or tree-of-thought where the LLM explores more than one branch of the tree. But we can address this by modifying Eqn. 4. For example, in majority voting, if the model must select an answer that occurs more than half the time, Eqn. 4 would have an appropriate summation over binomial coefficients. This does not lead to a closed-form analytical formula like Eqn 7, but it can be calculated easily numerically. For tree-of-thought, one could determine the scaling law for the error of the next predicted “thoughts” given the heuristic judge and obtain similar results—indeed the tree of thought is a tree. We will include this comment (and our comments from FAQ 4 in the Appendix on this issue) into the final paper.
>
>
> > **...optimal reasoning depth prediction on larger, frontier LLMs, especially those like DeepSeek-R1 that are known for long reasoning paths**
>
> We repeated the experiment in Fig 9 (right) with DeepSeek-V3 on GSM8k and MATH-500 (https://imgur.com/a/MYEFjEj). The results are interesting. For GSM8k, the error plateaus at small CoT lengths and increases thereafter. For MATH-500, the error is high at small CoT lengths and decreases thereafter. Error of DeepSeek-V3 is lower than that of models like Qwen and Gemma in both cases (https://imgur.com/a/MYEFjEj, and Fig 9 right). This is actually consistent with our theory. GSM8k has easier questions and it is more likely that the optimal reasoning length is much smaller than the typical CoT length; in other words, the left part of the parabola in Fig 9 (right) is not sampled in the dataset. MATH-500 has harder questions and so the reverse is true. Furthermore, Wu et al., 2025 provide extensive empirical evidence for an optimal reasoning length which we cite.

---

> > ### Author Rebuttal · Reviewer_4Bg9 · 2026-04-03
> >
> > My concerns have been adequately addressed

---

> > > ### Author Response · Authors · 2026-04-03
> > >
> > > Thank you for your feedback. We are glad that our response has addressed all the Reviewer’s concerns. We would be grateful if you would increase your score and champion our paper. Please let us know if you would like further clarifications on any other aspects of the manuscript. We would be glad to engage in a discussion.

---

### Official Review · Reviewer_NRhV · 2026-03-13

**Soundness:** 3
**Presentation:** 3
**Significance:** 3
**Originality:** 3
**Overall Recommendation:** 3
**Confidence:** 4

**Summary:**

The paper develops a theoretical framework for understanding when Chain-of-Thought (CoT) reasoning improves prediction accuracy in large language models. The authors model reasoning tasks as tree-structured classification problems, where each reasoning step corresponds to selecting among plausible next tokens. Using assumptions from statistical learning theory, the paper derives scaling laws for classification error with respect to the number of classes and intrinsic dimensionality of the input space. The analysis predicts that CoT is beneficial when reasoning decomposes a large classification problem into balanced intermediate steps and that there exists both an optimal branching factor and optimal reasoning depth. The authors further argue that increasing reasoning length beyond this optimal point may harm accuracy (‘overthinking’). These predictions are supported by synthetic experiments and limited empirical evaluation on GSM8K using several LLMs. Overall, the paper outlines an important concept by proposing a theoretical explanation for when reasoning improves or degrades performance.

**Compliance With Llm Reviewing Policy:**

Affirmed.

**Final Justification:**

I appreciate the rebuttals but many concerns remained so I maintain my score.

**Key Questions For Authors:**

•	The experiments focus primarily on synthetic tasks and GSM8K. Would the theoretical predictions hold on other reasoning benchmarks such as MATH, ARC, or code generation tasks?

•	The theory assumes reasoning trees with identifiable branching factors. Have you attempted to empirically estimate the branching factor of reasoning traces generated by LLMs?

•	The analysis relies on intrinsic dimensionality estimates of latent representations. How sensitive are the results to the method used to estimate this dimensionality?


•	Many modern reasoning methods use search-based strategies (e.g., self-consistency, tree-of-thought). How would these methods fit within the proposed theoretical framework?

•	Would the theoretical predictions change for models trained explicitly with reinforcement learning or reasoning supervision compared to standard language models?

**Limitations:**

The authors discuss limitations related to the assumptions of the reasoning-tree model and the applicability primarily to convergent reasoning tasks.

**Strengths And Weaknesses:**

Strengths

•	The paper introduces a clear conceptual model that represents reasoning tasks as tree-structured classification problems. This abstraction allows the authors to analyze CoT reasoning using tools from statistical learning theory and neural scaling laws. The framework offers an intuitive explanation for why decomposing complex reasoning into intermediate steps can improve accuracy.

•	The theoretical analysis leads to several predictions, namely, (1) reasoning trees with balanced branching factors minimize error (2) there exists an optimal branching factor (3) there exists an optimal reasoning depth and (4) excessive reasoning can degrade performance. These predictions provide a structured explanation for empirical phenomena such as the ‘overthinking’ effect reported in recent LLM literature.

•	The derivation linking classification error to the number of classes through a power-law relationship provides a plausible connection between reasoning decomposition and known scaling behaviors in deep learning.

•	The paper attempts to support its theoretical claims through synthetic reasoning experiments and empirical observations on GSM8K. While limited, this combination of theory and experimentation helps ground the analysis.

Weaknesses

•	The analysis relies on several strong assumptions, including (a) reasoning tasks can be represented as balanced trees (b) reasoning steps correspond to discrete classification problems with fixed branching factors (c) intrinsic dimensionality of the model representation remains constant across reasoning steps and (d) Bayes error is unaffected by reasoning decomposition.  These assumptions significantly simplify the reasoning process and may not accurately reflect real LLM behavior.

•	The empirical evaluation is relatively narrow. Most results are demonstrated on synthetic reasoning tasks, with only limited real-world validation (e.g., GSM8K). Given the central theoretical claims, broader experiments on additional reasoning benchmarks (such as MATH, ARC, or coding tasks) would strengthen the evidence.

•	Modeling reasoning as a fixed tree structure may overlook important aspects of reasoning in language models, such as backtracking and correction, latent reasoning not reflected in tokens and dynamic branching behavior during generation. These phenomena are common in current reasoning models and may violate the assumptions of the theoretical framework.

•	The paper occasionally suggests that optimal reasoning depth is a general property of LLM reasoning. However, the experiments provided are limited and do not fully establish the universality of this phenomenon.

•	Several recent works attribute reasoning performance improvements to factors such as training data structure, verification signals, or search strategies. The paper would benefit from more extensive discussion of how its theoretical framework relates to these explanations.

---

> ### Author Rebuttal · Authors · 2026-03-31
>
> Thank you for the feedback. We appreciate that you found our model clear and grounded. We have addressed some of your comments below.
>
> > **The analysis relies on several strong assumptions… may not reflect real LLM behavior.**
>
> > **How sensitive are the results to the method used to estimate this dimensionality?**
>
> Consider a task where an LLM is given a prompt and it selects an answer from many possible choices. This is, in effect, a classification problem. The LLM could either produce the answer directly, or it could identify the answer using a sequence of steps; each step corresponding to a classification sub-task that extends the context for the next step. The associated text is called a chain of thought. By construction, chain-of-thought has a tree-like structure. Thinking is a concatenation of multiple chains of thought, and therefore it also has a tree-like structure. Therefore, the tree structure is not an assumption.
>
> We show that the probability of error in classification problems with a large number of output classes can be reduced by decomposing the problem into a sequence of smaller ones. The error is minimized when the tree has equal degree at each level. These statements hold in general for any classification task. They are not assumptions.
>
> In an LLM, the inputs are sentences/contexts. We want to estimate the dimension of these sentences. Although each token is represented as a vector in some, say, $p$-dimensional space, the set of such vectors that make up a sentence of length $n$ may lie within a subspace of lower dimensionality---which we will call the intrinsic dimension $d$. Our experiments show that $d$ is independent of the reasoning step. In Fig 4 (left) we estimated $d$ as the number of principal components required to capture 80% of the variance of the log-probabilities over vocabulary (logits) used to select the next token from the context. We have tested other dimensionality reduction methods (https://imgur.com/a/MYEFjEj), including maximum likelihood estimate (Levina-Bickel, 2005), two-nearest-neighbors (Facco et al., 2017), Isomap (Tenebaum et al., 2000), and Kernel PCA with a radial-basis-function kernel. All of these methods showed a stable dimension with respect to context-length, indicating that our result is not specific to using PCA for dimensionality reduction.
>
> The Bayes error refers to the irreducible uncertainty of a task, independent of the strategy used to solve it. Bayes error does not depend on whether or not a model is using CoT.
>
> > **limited real-world evaluation (only GSM8k), more experiments on MATH, ARC etc**
>
> We conducted new  experiments on the MATH-500 dataset using Qwen2.5-7B-Instruct and Gemma3-1b-it (https://imgur.com/a/MYEFjEj). These results are similar to those in Fig. 9 (right). This shows that our theoretical predictions hold for a broader set of datasets. In fact, the work of Wu et al., 2025 which we have cited in our paper, provides extensive corroborating empirical evidence for our theory (see their Fig 1a). Please also see our response to the last question by Reviewer 4Bg9 for new experiments with DeepSeek.
>
> > **fixed tree structure may overlook other search-based strategies, e.g., self-consistency, tree-of-thought**
>
> Please see our response to a similar question by Reviewer 4Bg9. We did not reproduce it here for lack of space.
>
> > ​​**Have you attempted to empirically estimate the branching factor in LLMs**
>
> Fig 5 does this exactly using data from our synthetic task. The branching factor of the tokenized text of the transformer is correlated with the true branching factor in our synthetic task. We did a similar analysis for GSM8k. It is difficult to obtain a dense enough sampling of real data to estimate the branching factor precisely. A very rough estimate (computed using the edit distance between pairs of answers) gives a degree of ~1.3. This is interesting because it corresponds to an entropy of ~0.25 which almost perfectly matches our result in Fig 4 (right). On synthetic data similar to that of Fig 1, where we can sample all branches, our rough estimate is close to the correct degree.
>
> > **Several recent works attribute reasoning performance improvements to factors such as training data structure, verification signals, or search strategies…**
>
> FAQ 1 in the Appendix discusses some practical implications of our results. One way to interpret our results is that training on data that has a tree-like structure could enable the LLM to crystallize a similar tree structure in its reasoning traces, and improve accuracy. Training on redundant trees like those in Sec 4 would further improve the LLM. Verification signals like those in RL have been shown to reduce reasoning depth towards its optimal value (Wu et al. 2025). This is consistent with our results in Fig 8 (left) and 9. Also see FAQ 5 in the Appendix where we discuss RL. We will add this comment, including the one above on search strategies, to the paper.

---

> > ### Author Rebuttal · Reviewer_NRhV · 2026-04-04
> >
> > Thank you for providing some details on my questions. However, there are still some unanswered questions
> >
> > (1)	While representing CoT as a tree is a reasonable abstraction, your proof that error is minimized relies on the tree having an equal degree at each level (a perfectly balanced tree). In real-world reasoning tasks, some steps are strictly deterministic (branching factor of 1) while others involve exploring a massive search space (high branching factor). How do your theoretical scaling laws hold up when the tree is highly unbalanced or irregular, which is the norm for real reasoning problems? Isn't assuming a balanced tree still a significant idealized assumption in your model?
> >
> > (2)	I was hoping you will use some newer MATH benchmarks like AIME. MATH-500 is heavily saturated and likely in the training sets of these newer base models, how did you control for data contamination?
> >
> > (3)	Your framework suggests that increasing reasoning length beyond an optimal point harms accuracy ("overthinking"). However, state-of-the-art large reasoning models (like OpenAI's o3) use massive reinforcement learning pipelines precisely to generate exceedingly long chains of thought (sometimes thousands of tokens) to solve complex problems like AIME or code generation. This empirical evidence shows that scaling test-time compute via longer chains consistently yields better answers on hard tasks. How does your framework reconcile the success of these long-thinking models with your theory that longer reasoning paths degrade accuracy?
> >
> >
> > I will stay with my previous rating.

---

> > > ### Author Response · Authors · 2026-04-06
> > >
> > > We appreciate these comments and welcome the opportunity to address them. We will incorporate these responses in the paper.
> > >
> > > **(1) and (3)** Our theory never assumes that the reasoning tree is balanced. Eqn 5 is the difference of the error by direct prediction and by reasoning, it is a function of the degrees $\{m_1, \dots, m_n\}$ of different depths $n$. As we show on Line 264–270, this difference is minimized when $m_k = m^* = e^{d/2}$ for all $k$. The balanced tree is therefore a result, not an assumption. We also empirically validate that error is minimized by a balanced tree using the synthetic task in Fig. 1.
> > >
> > > Next, Wang et al. (2025) observed in [1], a very small number of reasoning steps generated by an LLM have high entropy and are important to the eventual accuracy. In other words, the reasoning traces of an LLM look like a tree where many steps have a very small degree (close to 1 in fact). This is not inconsistent with our theory which suggests that if the reasoning tree has an equal degree at every level, extending reasoning depth can be detrimental. The fact that reasoning traces in LLMs are exceedingly long simply suggests—in light of our theory—that they might not be efficient even if they are accurate. If one constructs shorter, balanced reasoning trees, we might obtain similar accuracies with fewer computational resources.
> > >
> > > Although there is a wide belief that long-thinking models are more accurate, the empirical evidence is not yet definitive. As Wu et al. (2025) also showed (cited in our paper), the accuracy of reasoning decreases with “overthinking”. While Deepseek does well with long reasoning traces today, a different model might do even better with shorter reasoning traces tomorrow. This is why it is important to test such predictions in controlled settings such as our synthetic experiment.
> > >
> > > [1] Wang et al., “Beyond the 80/20 Rule: High-Entropy Minority Tokens Drive Effective Reinforcement Learning for LLM Reasoning” (2025)
> > >
> > > **(2)** Thanks for the suggestion. We ran Deepseek-v3 on the AIME dataset (https://imgur.com/a/fdPEd5b).  AIME contains diverse problems of very different difficulty, and therefore there may not be a single optimal reasoning length for the entire dataset. This is why we conducted this analysis by splitting AIME into four subsets, ranging from questions with the smallest reasoning length, to those with the largest reasoning length. As the figure shows, the error is a non-monotonic function of the reasoning length. For all subsets, the error deteriorates with reasoning length after a minimum at an intermediate value. This corroborates the existing experimental results in our paper.
> > >
> > > That said, data contamination is an issue in many typical evaluations of LLMs. It can indeed be a factor in the empirical validation of our theory. AIME is a famous high school mathematics competition and it is likely that new LLMs contain data from even AIME in their training dataset. The empirical results of Wu et al., who we also cite in our paper, are also consistent with our empirical observations and our theory—the error does not necessarily decrease as the reasoning length increases.

---

### Official Review · Reviewer_tBS6 · 2026-03-20

**Soundness:** 3
**Presentation:** 4
**Significance:** 3
**Originality:** 4
**Overall Recommendation:** 5
**Confidence:** 2

**Summary:**

This paper develops a theoretical framework for understanding when and why Chain-of-Thought (CoT) reasoning improves or hurts performance in LMs, modeling reasoning tasks as trees where the root is a prompt and leaves are answers. The authors theoretically prove that CoT helps reduce the error and validate their findings with experiments.

Authors show that CoT is most effective when the reasoning tree has a roughly equal branching degree at each level, since this maximally structured decomposition maximizes the accuracy gain over direct prediction.

They then extend this analysis to increasing reasoning depth by introducing redundant paths leading to the same answer, and show that increasing depth without increasing the degree does not lead to fewer errors, and there is an upper bound on depth beyond which it cannot help further.

The paper also shows that larger models, having higher intrinsic dimensionality, have an optimal reasoning depth that is shorter, which is consistent with empirical findings that stronger models benefit from more concise reasoning traces. These predictions are validated on synthetic logical deduction tasks and real benchmarks like GSM8k using Qwen and Gemma models.

**Compliance With Llm Reviewing Policy:**

Affirmed.

**Key Questions For Authors:**

--

**Limitations:**

yes

**Strengths And Weaknesses:**

# Strengths

* Tackles an important problem of understanding when reasoning works and does not.
* Good theoretical framework and well-designed experiments to back it up empirically.
* The findings should be useful for the broader NLP/AI community, as reasoning is standard in LLMs now.
* The paper is very well written.

# Weaknesses

It's unclear to me how the insights derived from this theoretical understanding could help improve reasoning models today. Some more discussion on this would have strengthened the paper.

---

> ### Author Rebuttal · Authors · 2026-03-31
>
> Thank you for your feedback. We are glad you found that the paper is well-written and that it develops a good theoretical framework. We included a response to your comment below.
>
> > **It's unclear to me how the insights derived from this theoretical understanding could help improve reasoning models today. Some more discussion on this would have strengthened the paper.**
>
> We agree that it is beneficial to discuss how our theoretical insights can be used to improve current models. One of the paragraphs in the discussion section talks about test-time scaling. Our results indicate that arbitrarily scaling up the thinking time of an LLM does not necessarily improve accuracy.
>
> In our FAQs section in the Appendix, we directly address your question: “Are there any practical guidelines for building real-world reasoning systems?”. An improved theoretical understanding of CoT can help steer future research towards ideas that are more likely to be successful. Our analysis also indicates that CoT does not have to be human-understandable. It is the shared structure of the reasoning traces that yields the benefit in accuracy. Thus, it should be possible to cheaply generate training data with a tree-like structure to solve reasoning problems without relying entirely on expensive human-generated reasoning traces. In fact, it may be possible to develop a training paradigm that automatically augments the training data with reasoning traces of optimal depth and degree. Finally, we expect that the same tree-like structure present in standard CoT can be used to decompose other classification tasks such as image classification, protein structure analysis, robotics, etc.
>
> We will move this text to the main paper.

---

### Decision · Program_Chairs · 2026-04-30

**Decision:**

Accept (regular)

**Comment:**

This paper develops a theoretical framework for understanding when Chain-of-Thought reasoning improves or degrades performance, modeling reasoning tasks as tree-structured classification problems and deriving scaling laws for prediction error with respect to branching degree and reasoning depth. The theoretical contributions are clear and novel, linking CoT to statistical learning theory and providing principled explanations for the "overthinking" phenomenon. The predictions are validated on both synthetic tasks and real benchmarks. However, empirical evaluation could be broader, the framework is restricted to convergent reasoning tasks, and the connection to practical system design remains indirect.